# SARS-CoV-2 infection results in immune responses in the respiratory tract and peripheral blood that suggest mechanisms of disease severity

Wuji Zhang [1], Brendon Y. Chua[1,2], Kevin J. Selva[1], Lukasz Kedzierski[1,3], Thomas M. Ashhurst [4,5], Ebene R. Haycroft [1], Suzanne K. Shoffner-Beck [6], Luca Hensen[1], David F. Boyd [7], Fiona James [8], Effie Mouhtouris[8], Jason C. Kwong[1,8], Kyra Y. L. Chua [8], George Drewett[8], Ana Copaescu[8], Julie E. Dobson[9], Louise C. Rowntree[1], Jennifer R. Habel [1], Lilith F. Allen[1], Hui-Fern Koay [1], Jessica A. Neil [1], Matthew J. Gartner[1], Christina Y. Lee [6], Patiyan Andersson[10], Sadid F. Khan[11], Luke Blakeway[11], Jessica Wisniewski [11], James H. McMahon[11,12], Erica E. Vine [13,14,15], Anthony L. Cunningham [13,15], Jennifer Audsley [16], Irani Thevarajan[16,17], Torsten Seemann[10], Norelle L. Sherry [8,10], Fatima Amanat [18,19], Florian Krammer [18], Sarah L. Londrigan [1], Linda M. Wakim [1], Nicholas J. C. King [4,5,14,20,21,22], Dale I. Godfrey [1], Laura K. Mackay [1], Paul G. Thomas [7], Suellen Nicholson [23], Kelly B. Arnold [6], Amy W. Chung [1], Natasha E. Holmes[8,24,25,26], Olivia C. Smibert[8,27,28], Jason A. Trubiano[26,27,28,29,30 ✉], Claire L. Gordon [1,8,30 ✉], Thi H. O. Nguyen [1,30 ✉] & Katherine Kedzierska [1,2,30 ✉]

Respiratory tract infection with SARS-CoV-2 results in varying immunopathology underlying COVID-19. We examine cellular, humoral and cytokine responses covering 382 immune components in longitudinal blood and respiratory samples from hospitalized COVID-19 patients. SARS-CoV-2-specific IgM, IgG, IgA are detected in respiratory tract and blood, however, receptor-binding domain (RBD)-specific IgM and IgG seroconversion is enhanced in respiratory specimens. SARS-CoV-2 neutralization activity in respiratory samples correlates with RBD-specific IgM and IgG levels. Cytokines/chemokines vary between respiratory samples and plasma, indicating that inflammation should be assessed in respiratory specimens to understand immunopathology. IFN-α2 and IL-12p70 in endotracheal aspirate and neutralization in sputum negatively correlate with duration of hospital stay. Diverse immune subsets are detected in respiratory samples, dominated by neutrophils. Importantly, dexamethasone treatment does not affect humoral responses in blood of COVID-19 patients. Our study unveils differential immune responses between respiratory samples and blood, and shows how drug therapy affects immune responses during COVID-19.

A full list of author affiliations appears at the end of the paper.

Symptoms of SARS-CoV-2 infection, known as coronavirus disease 2019 (COVID-19), vary from asymptomatic or mild disease to critical illness, including respiratory failure and death[1]. Global efforts focused on developing new drugs and vaccines. While vaccines showed safety and immunogenicity towards SARS-CoV-2[2–4], the effects of drug treatments remain controversial. Dexamethasone, a synthetic glucocorticoid drug, can lower the 28-day mortality rate in COVID-19 patients receiving oxygen support, prolong ventilator-free days and improve oxygen partial pressure to fractional inspired oxygen ($PaO_2/FiO_2$) ratios compared to placebo or standard care[5–7]. However, SARS-CoV-2 RNA can be detected for longer in patients receiving glucocorticoid treatment[8]. Treatment with remdesivir, a nucleoside analogue inhibiting RNA-dependent RNA polymerase (RdRp), in COVID-19 can shorten the time to recovery and provide better clinical outcomes[9–11]. However, the effects of dexamethasone and/or remdesivir on humoral and cellular immune responses are unclear.

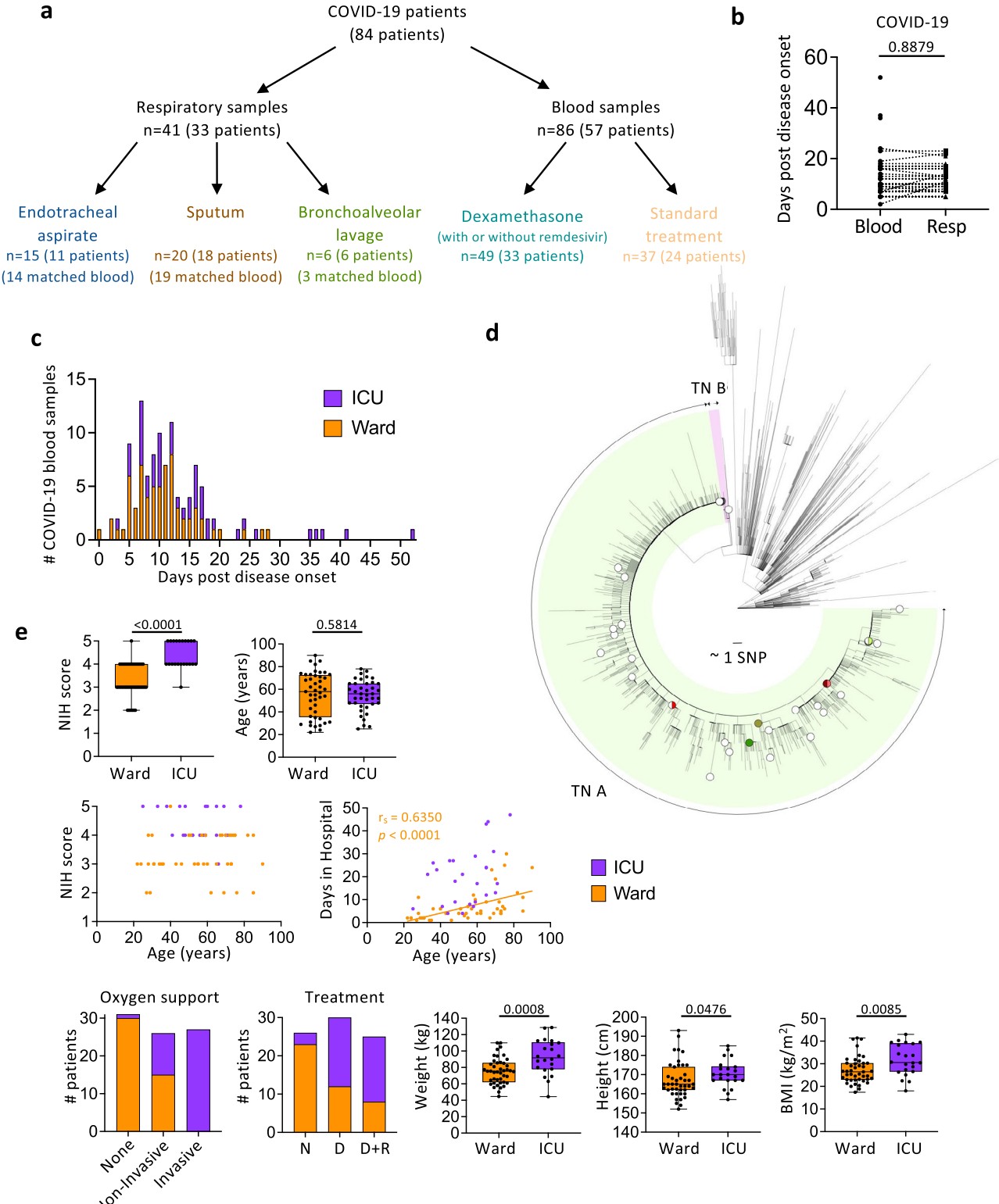

**Fig. 1 Demographics of the COVID-19 cohort. a** Number of patients recruited in the COVID-19 cohort and samples collected are shown. **b** Comparison between the time of respiratory sample (endotracheal tube aspirate (ETA), sputum, and bronchoalveolar lavage (BAL)) and paired blood sample collection are shown and were calculated using a two-sided Mann-Whitney test. $n_{Respiratory} = 40$, $n_{Blood} = 33$. **c** Collection time of COVID-19 blood samples. **d** Maximum likelihood phylogenetic tree of SARS-CoV-2 sequences from Victoria from 28 January 2020 to 28 October 2020 (including context sequences from the rest of Australia and New Zealand). Phylogenetic tree includes randomly subsampled sequences from transmission networks (TN) A and TN B in Victoria, with a total number of 10941 and 145 cases respectively. The outermost tip of each radial line represents a single sequence; the sum of each radial line between two tips represents the genetic distance between two sequences. Each radial stepwise progression represents approximately one single nucleotide polymorphism (SNP). Sequences from study patients ($n = 40$) are shown as open circles (patient with blood samples only) or solid-coloured circles (patients with blood and respiratory samples). Half-filled circles are used when samples are located close to each other. **e** Distribution of clinical data in ward and intensive care unit (ICU) COVID-19 patients. $n_{Ward} = 45$, $n_{ICU} = 39$. The patients received dexamethasone (D), dexamethasone with remdesivir (D + R) or neither (N). The bounds of the box plot indicate the 25th and 75th percentiles, the bar indicates medians, and the whiskers indicate minima and maxima. Statistical significance was determined with a two-sided Fisher's exact test for the National Institutes of Health (NIH) score, and a two-sided Mann-Whitney test for age, weight, height, and body weight index (BMI). Correlation was determined with a two-tailed Spearman's correlation. Source data are provided as a Source Data file.

Immunity towards SARS-CoV-2 infection has been studied, predominantly in peripheral blood. While transient, robust and broad immune responses precede patients' recovery in non-severe cases[12–15], severe COVID-19 can be associated with exuberant cytokine responses, hyperactivation of innate immune cells, reduced T cell numbers[12,14,16,17] and high titres of SARS-CoV-2-specific antibodies[16]. In contrast, immune responses in the respiratory tract are understudied. High levels of IL-6, IL-10, monocyte chemoattractant protein (MCP)-1, macrophage inflammatory protein (MIP)-1α and MIP-1β are detected in bronchoalveolar lavage fluid (BALF) of COVID-19 patients, indicating inflammatory environment with high monocyte chemoattractants[18,19]. While IFN-α and IFN-β were undetectable, IL-10, IL-17A and IL-18 were variably detected in COVID-19 BALF, with higher RNA and/or protein levels of IL-6, MCP-1 and IL-33 and lower IL-6 receptor (IL-6R) observed in COVID-19 BALF compared to healthy BALF[19,20]. Granulocytes and monocytes/macrophages dominate in COVID-19 airways, especially intermediate (CD14$^+$CD16$^+$) and non-classical (CD14$^-$CD16$^+$) monocytes[18,21]. Conversely, low frequencies of T cells were detected in COVID-19 airways with increased expression of activation markers CD38/HLA-DR and a tissue-resident phenotype[18,19]. Increased frequency of activated T cells in the airway is associated with improved survival[18].

To dissect the breadth of immune responses during SARS-CoV-2 infection in the respiratory tract compared to those detected in blood, we collected paired longitudinal blood and respiratory samples from hospitalised COVID-19 patients to investigate innate, adaptive and humoral immunity. Overall, our study unveils differences and defines correlations in innate and adaptive immune responses between respiratory and blood samples of COVID-19 patients and provides insights into potential biomarkers and immunotherapies for severe COVID-19.

## Results

**COVID-19 patient cohort**. To define immune responses to SARS-CoV-2 in the respiratory tract, we obtained 41 respiratory samples (15 endotracheal aspirates (ETA; from 11 patients), 20 sputum samples (from 18 patients), 6 bronchoalveolar lavage samples (BAL; from 6 patients)). Respiratory samples were collected from 33 PCR-positive COVID-19 patients from whom we also collected 34 paired blood samples (Fig. 1a, Supplementary Table 1 and Supplementary Table 2). Three COVID-19 patients were admitted to the ward while 30 patients were in the ICU (Fig. 1a; Supplementary Table 1). The median age of COVID-19 patients from whom we obtained respiratory samples was 55 years (range 25–76) and 33.3% were female (Supplementary Table 1). Where feasible, blood was collected on hospital

admission, during hospital stay and on hospital discharge. No significant differences were found between time of respiratory specimen collection and matched blood samples collected at the closest time-point ($p = 0.89$; Fig. 1b).

To determine the effects of dexamethasone, an anti-inflammatory corticosteroid, taken alone or in combination with the anti-viral drug remdesivir, on immune responses in blood, we recruited 57 COVID-19 patients (42 ward patients and 15 ICU patients) with a median age of 58 years (range 22–90) and 49.1% female from whom we obtained 86 blood samples (Fig. 1a, Supplementary Table 1, Supplementary Table 2).

SARS-CoV-2 genome sequence data, available from 40 out of 84 COVID-19 patients, showed that patients were infected with SARS-CoV-2 viruses belonging to the transmission network (TN)-A, representing a highly clonal and dominant network during the second wave of COVID-19 epidemic in Victoria[22], except for 1 patient belonging to TN-B (#001) (Fig. 1d).

**ICU admission associated with higher NIH severity score, oxygen therapy, drug treatment and weight**. Disease severity within our cohort was stratified according to whether COVID-19 patients were hospitalized in the ward or ICU. COVID-19 patients were also graded according to the NIH severity score of 1–5 according to their symptoms (Supplementary Table 3). ICU patients had significantly higher NIH scores compared to ward patients ($p < 0.0001$, Fig. 1e) and more ICU patients received oxygen support ($p < 0.0001$) and drug treatments, either dexamethasone alone or dexamethasone with remdesivir ($p < 0.0001$) (Fig. 1e, Supplementary Table 2). Interestingly, ICU patients also had significantly increased body weight ($p = 0.0008$), height ($p = 0.0476$) and body mass index (BMI) ($p = 0.0085$). Age correlated with the length of hospital stay ($p < 0.0001$, Fig. 1e), but no differences in age, gender, ethnicity, immunosuppressant drugs or smoking were observed between ICU and ward patients (Fig. 1e, Supplementary Table 2).

**Differential inflammatory cytokine profiles in respiratory and plasma samples**. There are scarce data on the inflammatory milieu in COVID-19 respiratory specimens. To determine cytokine/chemokine levels and composition in respiratory samples compared to paired plasma, we measured cytokines/chemokines (IL-1β, IFN-α2, IFN-γ, TNF, MCP-1, IL-6, IL-8, IL-10, IL-12p70, IL-17A, IL-18, IL-23 and IL-33), sIL-6Rα and an extracellular matrix protein "disintegrin and metalloproteinase with thrombospondin motifs-4" (ADAMTS4)[23]. Amongst the COVID-19 patients, greatly elevated levels of inflammatory cytokines/chemokines were detected in respiratory samples across ETA, sputum and BAL specimens, with concentrations being 160× (MCP-1), 90× (IL-6) and 110× (IL-8)

higher than in plasma (Fig. 2a). While IL-18 dominated in plasma, IL-6, IL-8 and MCP-1 were most prevalent in respiratory samples in patients with high cytokines/chemokines (Fig. 2a, Supplementary Fig. 1a). MCP-1, IL-6 and IL-8 were significantly higher in sputum and BAL than in plasma ($p < 0.0001$ and $p = 0.0476$; higher median also observed in ETA, though not significant), IFN-γ, IL-12p70,

IL-17A, IL-23 and IL-33 were significantly higher in plasma than in respiratory samples ($p < 0.0001$-$p = 0.0484$; Fig. 2b). IL-1β and IL-18 were also higher in sputum but not in ETA or BAL than in plasma ($p < 0.0001$ and $p = 0.0023$ respectively; Fig. 2b). In contrast, concentrations of IFN-α2, IL-10 and TNF were comparable across respiratory and plasma specimens, while sIL-6Rα was lower in

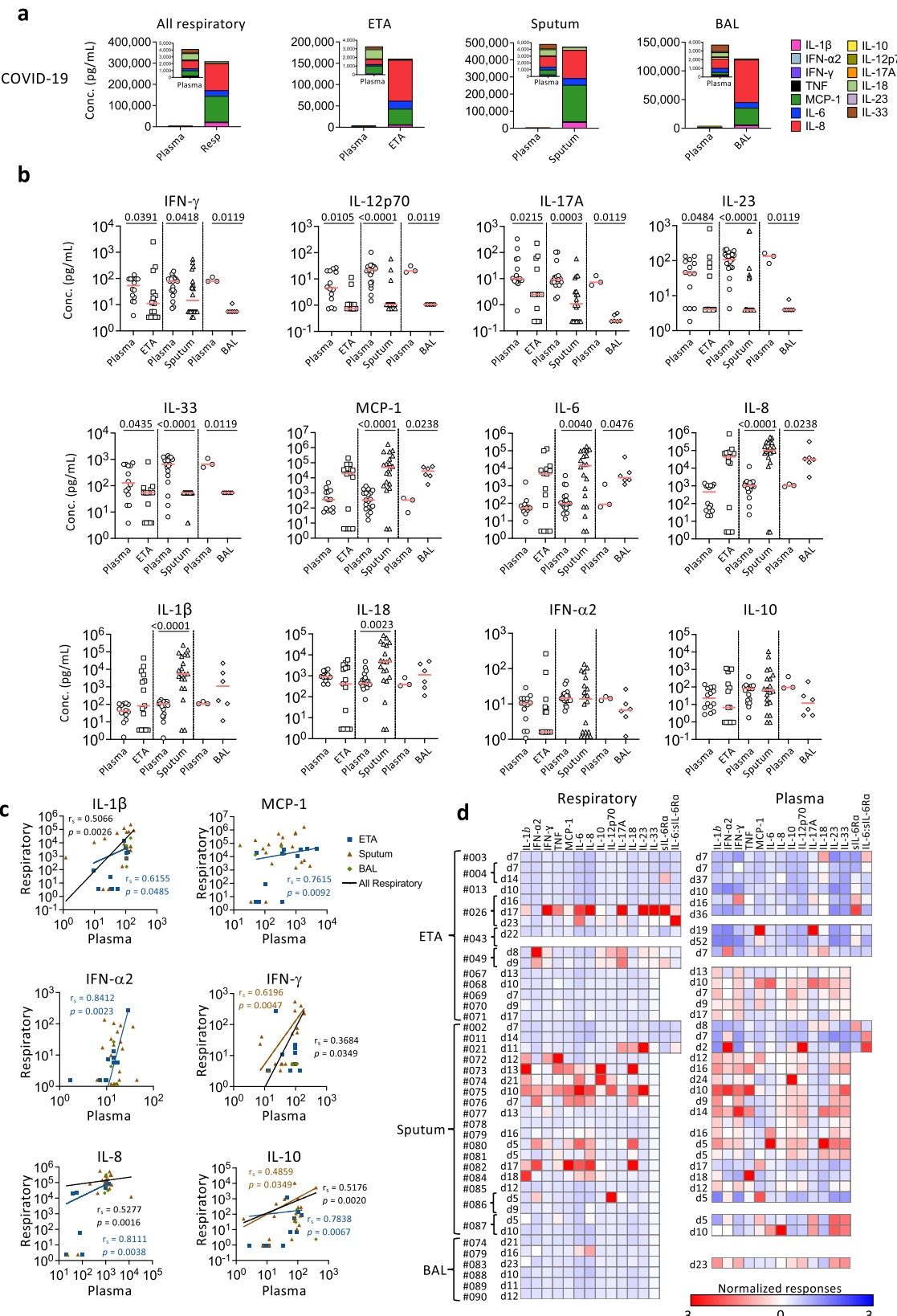

**Fig. 2 Discordant levels of cytokines and chemokines in COVID-19 respiratory samples compared to paired plasma samples. a** Absolute concentrations of 13 cytokines and chemokines (IL-1β, IFN-α2, IFN-γ, TNF, MCP-1, IL-6, IL-8, IL-10, IL-12p70, IL-17A, IL-18, IL-23, and IL-33) in pooled respiratory and paired plasma samples. **b** Comparison of cytokine and chemokine levels between endotracheal tube aspirate (ETA), sputum or bronchoalveolar lavage (BAL) and paired plasma samples using a two-sided Mann-Whitney test. Bars indicate the median values. **c** Correlation of cytokine and chemokine levels between respiratory samples (ETA, sputum, and BAL) and paired plasma samples collected at the closest timepoint for each patient. Correlation was determined with a two-tailed Spearman's correlation. $n_{ETA} = 15$, $n_{ETA\ matched\ plasma} = 14$, $n_{Sputum} = 20$, $n_{Sputum\ matched\ plasma} = 19$, $n_{BAL} = 6$, $n_{BAL\ matched\ plasma} = 3$. **d** Normalized levels of cytokines/chemokines for COVID-19 respiratory and plasma samples separately. Red color indicates higher cytokine/chemokine levels. Source data are provided as a Source Data file.

respiratory specimens than in plasma (Fig. 2b, Supplementary Fig. 1b). Furthermore, we found significant correlations between respiratory samples and plasma with respect to IL-1β, MCP-1, IFNα2, IFNγ, IL-8 and IL-10 (Fig. 2c). It is important to note that the 13 cytokines/chemokines measured were not detected in ETA or sputum samples of five COVID-19 ICU patients (#002, #003, #004, #011, #013), while high IL-18 levels were detected in the plasma of patients #002, #003, #004 and #013 (Supplementary Fig. 1a), demonstrating that cytokine levels can vary across both COVID-19 patients in respiratory samples and between paired respiratory and plasma samples.

Since the magnitude of cytokines/chemokines was much higher in respiratory samples than in plasma, a z-score normalization was performed for respiratory and matched plasma samples separately (Fig. 2d). High normalized cytokine levels were detected in ETA samples from only 2 out of 11 COVID-19 patients (#26 and #49) (Fig. 2d). In sputum samples, however, this was seen in 10 out of 18 patients (i.e. higher than ETA and BAL samples; #21, #72, #73, #74, #75, #76, #80, #81, #82 and #84). Our data therefore suggest that sputum potentially represents the more desirable specimen type that reflects the high inflammatory milieu at the primary site of SARS-CoV-2 infection. Conversely, the majority of patients displayed elevated cytokine/chemokine levels within plasma samples. Overall, while the inflammatory cytokine/chemokine levels were excessively higher in respiratory fluid compared to plasma, they were variable across COVID-19 patients, indicating that the plasma inflammatory milieu does not always reflect the airway inflammation and that hospitalized/ICU COVID-19 patients should be monitored for inflammation in airways, such as in sputum, to understand disease severity and potential benefits of immunomodulatory treatments.

**High RBD-specific IgM and IgG seroconversion in COVID-19 respiratory samples.** SARS-CoV-2-specific antibodies in respiratory samples are relatively unexplored. We measured SARS-CoV-2 RBD-specific IgM, IgG and IgA antibodies in paired respiratory and blood samples using RBD-ELISA and surrogate virus neutralisation test (sVNT) (Fig. 3). Compared to non-COVID-19, COVID-19 patients displayed higher levels of RBD IgM ($p = 0.0003$) and IgG ($p < 0.0001$), but not IgA, in respiratory samples (Fig. 3a, b), which was possibly either due to technical issues or cross-reactivity of IgA antibodies (Fig. 3b top left panel). However, significantly lower titres of RBD IgM and IgG were found in COVID-19 respiratory samples compared to matched plasma samples (Fig. 3b top right panel). This was consistent for both pooled respiratory samples (Fig. 3b top right panel) as well as separately analysed ETA, sputum and BAL samples (Fig. 3b bottom panel), with the exception of ETA IgG titres which had a lower median than matched plasma samples, though not significant.

Using sVNT, more sputum than ETA or BAL samples had detectable neutralizing activity, associated with high levels of RBD-specific IgM and IgG antibodies (Fig. 3c, d). Neutralizing activity was not detected in the majority (58.5%) of respiratory samples at the acute time-points. This included 11 ETA,

11 sputum and 2 BAL samples. Plasma samples with high neutralizing activity had high levels of all three Ig isotypes of RBD-specific antibodies, which positively correlated with neutralizing activity (Fig. 3c, d). This was also observed in respiratory samples, though only anti-RBD IgM and IgG significantly correlated with neutralizing activity (Fig. 3c, d). Seroconversion levels of RBD-specific IgM and IgG antibodies were detected in the majority of COVID-19 respiratory samples (34/41, 83%) and patients (26/33, 79%) (Fig. 3e), suggesting the prominence of RBD-specific IgM and IgG in respiratory samples during acute COVID-19. In terms of correlations between respiratory samples and plasma, overall IgM, IgG and sVNT levels correlated across the specimens (Fig. 3f).

**High prevalence of SARS-CoV-2-specific IgM and IgG antibodies in respiratory samples.** While anti-RBD antibodies are essential for the neutralization of SARS-CoV-2[24], non-neutralizing antibodies also have an important role in antiviral immunity[25]. To understand in-depth antibody profiles and cross-reactivity in respiratory samples, we adapted a multiplex bead array assay[25] (Supplementary Table 5). Antibodies targeting RBD, S proteins and NP of SARS-CoV-2, SARS-CoV-1 and human coronaviruses (229E, NL63, OC43, HKU1) were assessed for isotype/subclass (IgM, IgG, IgG1-4, IgA1-2) and binding with FcγR (FcγR2aH, FcγR2aR, FcγR2b, FcγR3aV, FcγR3aF) and C1q, totaling 315 features, in 14 COVID-19 and 5 non-COVID-19 respiratory samples and paired plasma. Intermediate to high antibody levels across different isotypes and SARS-CoV-2 antigens were detected in a subset of COVID-19 respiratory and plasma samples (Fig. 4a), especially in patients who lacked inflammatory cytokines in their respiratory samples (patients #002, #003, #004, #011, #013). Conversely, patients with low anti-SARS-CoV-2 antibodies (patients #021, #043, #049) could still have variable antibody responses towards other human coronavirus in plasma and/or respiratory samples (Supplementary Fig. 2a).

When comparing COVID-19 and non-COVID-19 respiratory samples, high levels of SARS-CoV-2-specific IgM, IgG, IgA1 and IgA2 were detected in COVID-19 (Fig. 4b). While low SARS-CoV-1 IgG and IgA2 levels were detected, no significant differences in antibodies against other human coronaviruses (229E, NL63, OC43, HKU1) were found (Fig. 4b). IgG1 and IgG3 were the most prominent subclasses (Supplementary Fig. 2b). SARS-CoV-2 antibodies with FcγR binding abilities were detected at low levels in COVID-19 respiratory samples (Supplementary Fig. 2b).

To investigate the most prominent antibody features that differed between COVID-19 and non-COVID-19 patients, Partial Least-Squares Discriminant Analysis (PLSDA) was performed (Fig. 4c). As few as 3 antibody features were sufficient to separate COVID-19 and non-COVID-19 ETA, with higher SARS-CoV-2-specific IgG and IgM in COVID-19 ETA, consistent with higher anti-RBD IgG and IgM in ELISAs (Figs. 3b, 4c). In contrast, higher SARS-CoV-2-S-specific IgG antibodies and antibodies

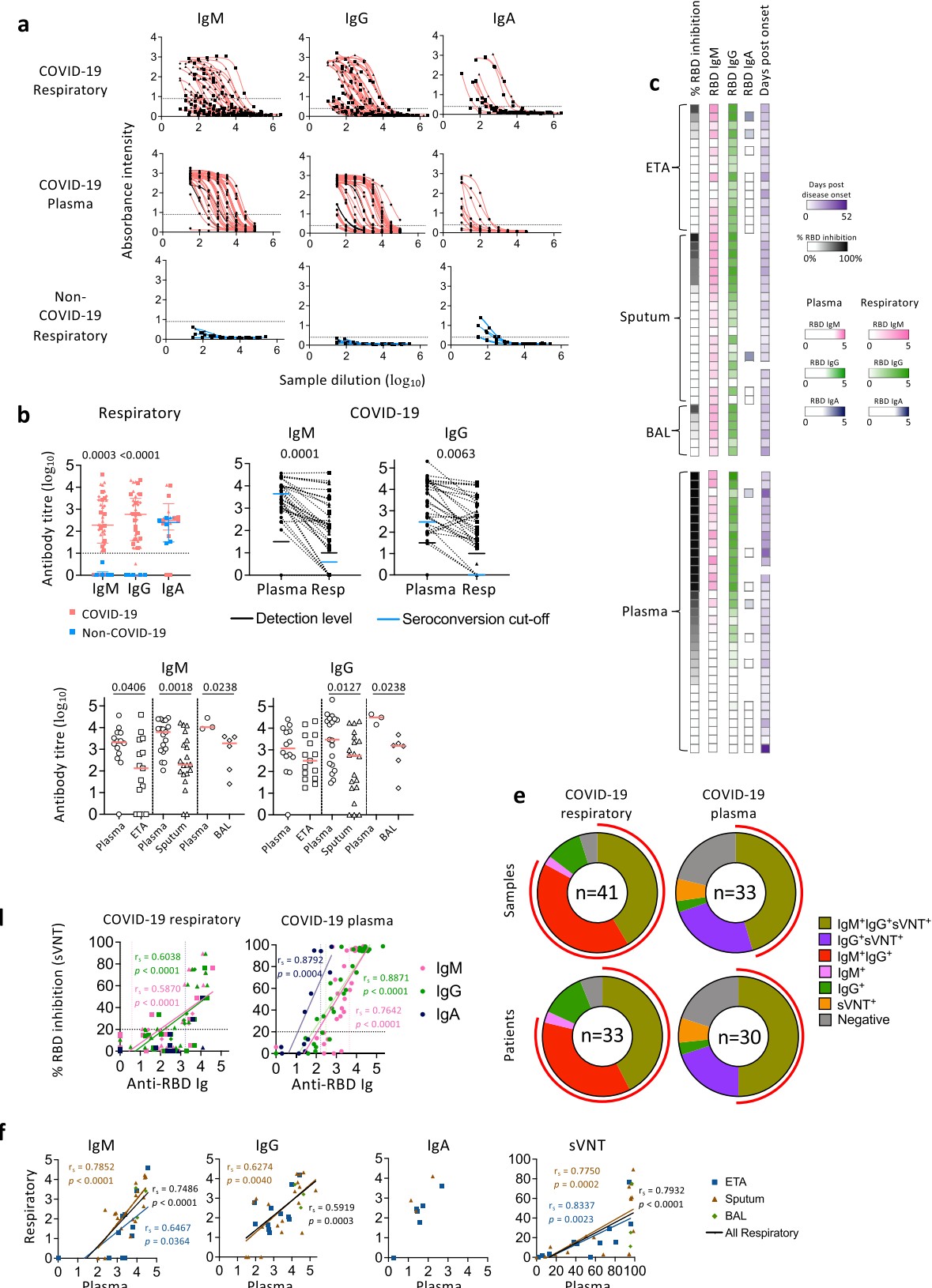

with FcγR binding activities were strongly featured in COVID-19 plasma compared to non-COVID-19 plasma (Fig. 4c).

**Increasing cellular infiltrates in respiratory specimens during disease progression**. To determine cellular immunity in

respiratory specimens of COVID-19 patients, samples underwent multi-parameter flow cytometry and analysis using the Spectre R package[26]. Cells were clustered using Flow Self-Organizing Map (FlowSOM)[27] and plotted using Fast Interpolation-based t-distributed Stochastic Neighbour Embedding (FIt-SNE)[28]. Two flow cytometry panels were used to ensure accurate profiling of

**Fig. 3 Higher anti-RBD IgM and IgG seroconversion rate in respiratory samples compared to paired plasma samples of COVID-19 patients. a** ELISA titration curves against the SARS-CoV-2 receptor-binding domain (RBD) for IgM, IgG, and IgA in COVID-19 respiratory and paired plasma samples and non-COVID-19 respiratory samples as negative controls. Dotted lines within each graph indicates the cut-off used to determine end-point titres. **b** End-point titres of SARS-CoV-2 RBD antibodies between top left panel: respiratory samples of COVID-19 and non-COVID-19 patients, top right panel: respiratory and paired plasma samples of COVID-19 patients, and bottom panel: endotracheal tube aspirate (ETA), sputum, or bronchoalveolar lavage (BAL) and paired plasma samples of COVID-19 patients. Top left panel: Bars indicate median with interquartile range. Dotted line indicates the detection level. $n_{ETA} = 15$, $n_{Sputum} = 20$, $n_{BAL} = 6$, $n_{Non-COVID-19\ ETA} = 5$, $n_{Non-COVID-19\ sputum} = 1$. Top right panel: Dotted lines connect the most closely matched plasma and respiratory samples from each patient. Bottom panel: Bars indicate the median. Statistical significance was determined with a two-sided Mann-Whitney test. $n_{ETA} = 15$, $n_{ETA\ matched\ plasma} = 14$, $n_{Sputum} = 20$, $n_{Sputum\ matched\ plasma} = 19$, $n_{BAL} = 6$, $n_{BAL\ matched\ plasma} = 3$. **c** Heatmap of percentage (%) inhibition tested by surrogate virus neutralization test (sVNT), anti-RBD ELISA titres and days post disease onset. **d** Correlation between anti-RBD antibody titres and (%) sVNT inhibition. Correlation was determined with a two-tailed Spearman's correlation. **e** Number of samples and patients with seroconverted anti-RBD IgM, IgG, IgA and positive % sVNT inhibition. Red curved lines surrounding the donut graphs indicate the samples/patients with seroconverted IgM and IgG. Earliest samples were used for each patient when determining seroconversion which was defined as average titre + 2×SD of non-COVID-19 respiratory samples. Positive % sVNT inhibition was defined as % sVNT inhibition ≥ 20%. **f** Correlation of anti-RBD ELISA titres and % sVNT inhibition between respiratory samples (ETA, sputum, and BAL) and paired plasma samples collected at the closest timepoint for each patient. Correlation was determined with a two-tailed Spearman's correlation. $n_{ETA} = 15$, $n_{ETA\ matched\ plasma} = 14$, $n_{Sputum} = 20$, $n_{Sputum\ matched\ plasma} = 19$, $n_{BAL} = 6$, $n_{BAL\ matched\ plasma} = 3$. Source data are provided as a Source Data file.

myeloid and lymphoid cell populations (Supplementary Fig. 3a, 4a, b, Supplementary Table 7).

Clustering of respiratory samples in the myeloid panel revealed that CD66b+neutrophils dominated, with varying levels of CD16 expression (Fig. 5a). CD14+macrophages and CD4+ and CD8+ T cells were also detected, but at lower frequencies. While the cellular component was variable across samples, CD16hi and CD16lo neutrophils were present in all COVID-19 patients apart from patient #043 (BMT recipient; Fig. 5a, Supplementary Fig. 4c). Although there were only two COVID-19 patients (patients #026, #049) with multiple ETA samples, we still observed an increase in cellular infiltrates over time, including CD16lo neutrophils (Fig. 5b). In the respiratory specimens of 6 non-COVID-19 patients, lower levels of neutrophils and macrophages were detected (Supplementary Fig. 4c). Non-COVID-19 patient #059 had a large population of CD16- neutrophils, while a high frequency of CD16lo neutrophils was detected in blood, indicating a dominant immature neutrophil population in this patient (Supplementary Fig. 4c, e).

After excluding neutrophils and monocytes/macrophages in respiratory samples, CD8+ T cells were the major population of lymphocytes, with varying levels of CD4+ T cells and natural killer (NK) cells (Fig. 5c, Supplementary Fig. 4d). Increasing infiltrates of T cells over time were found in patients #026 and #049 (Fig. 5d), similar to neutrophils. Interestingly, in patient #026, the lymphocyte population was dominated by NK cells early (d16 and d17) and T cells gradually infiltrated and dominated overtime (d23). Low lymphocyte levels were detected in fatal patient #021.

A volcano plot was generated to determine fold differences in immunological features between respiratory and blood samples (Fig. 5e, f). While cell numbers were higher in blood, higher frequencies of intermediate (CD14+CD16+) monocytes/macrophages, activated (HLADR+CD38+) and EM-like (CD27-CD45RA-) CD4+ and CD8+ T cells were found in COVID-19 respiratory specimens compared to blood. Respiratory specimens had a higher neutrophil to T cell ratio (Fig. 5f). Conversely, the ratio of CD4+ to CD8+ T cells was lower in respiratory samples ($p = 0.0065$), indicating a high prevalence of CD8+ T cells in respiratory specimens (Fig. 5f).

Overall, neutrophils (CD16+/−) dominated in the respiratory samples of COVID-19 patients, with varying levels of monocytes/macrophages, T cells (CD4+ and CD8+), NK cells, and B cells. T cells in the respiratory samples exhibited an activated and EM-like phenotype compared to paired blood samples, with lower CD4+ to CD8+ T cell ratios.

**IFN-α2 and IL-12p70 levels in ETA and RBD neutralizing activity in sputum negatively correlate with days of hospital stay.** To understand associations between clinical features and serological responses in the respiratory specimens, correlations between clinical data (age, weight, height, BMI, days post disease onset, days of hospital stay) and serological features (cytokines and chemokines, sIL-6Rα, ADAMTS4, anti-RBD IgM, IgG, IgA and sVNT inhibition) were performed for ETA and sputum samples separately (Fig. 6a–d). IFN-α2 and IL-12p70 levels in ETA negatively correlated with days of hospital stay, albeit there were low levels of IL-12p70 in respiratory samples (Fig. 6a, b). In sputum, sVNT inhibition activity negatively correlated with days of hospital stay, but positively correlated with levels of MCP-1, IL-6, IL-8 and IL-10 (Fig. 6c, d).

Regarding associations between immunological features in the respiratory specimens, correlations between serological features (cytokines, sIL-6Rα, ADAMTS4, anti-RBD IgM, IgG and IgA) and cellular features were performed (Fig. 6e, f). Lower frequency of classical (CD14+CD16−) monocytes and higher intermediate (CD14+CD16+) monocytes correlated with higher FcγR2b SARS-CoV-2-Trimer-S-specific antibodies ($p = 0.0005$; $p = 0.0004$). Higher SARS-CoV-2-specific antibodies correlated with frequencies of CM-like (CD27+CD45RA−) and CD16lo neutrophils ($p = 0.0034$; $p = 0.0076$). Numbers of intermediate (CD14+CD16+) monocytes, HLA-DR+ NK cells, CD8+ T cells (CM-like; EM-like) and CD16hi neutrophils correlated with antibody levels ($p = 0.0009$-$0.0039$). Unsupervised clustering also revealed distinct immunological features between respiratory and blood samples, with higher EM-like CD4+ and CD8+ T cell frequencies and lower cell numbers (apart from neutrophils) in the respiratory samples (Fig. 6g).

**COVID-19 patients with higher NIH scores had more robust humoral immune responses in blood.** While immune responses in blood samples between ward and ICU patients has been investigated in many studies, the classification of patients using NIH scores based on symptoms might correlate better with their immune responses. Unsurprisingly, more patients with higher NIH scores of 4-5 required ICU during hospitalization (Fig. 1d), while NIH scores of 2-3 were in the mild/moderate group. While all blood samples were collected during the acute phase of the infection, samples were grouped into hospital admission (V1) and hospital discharge (V7) for analyses. Although there were no differences in the overall cytokine/chemokine levels between the two NIH severity groups, IL-8 levels in the severe/critical group increased at V7 compared to V1 ($p = 0.0004$; Fig. 7a), indicating

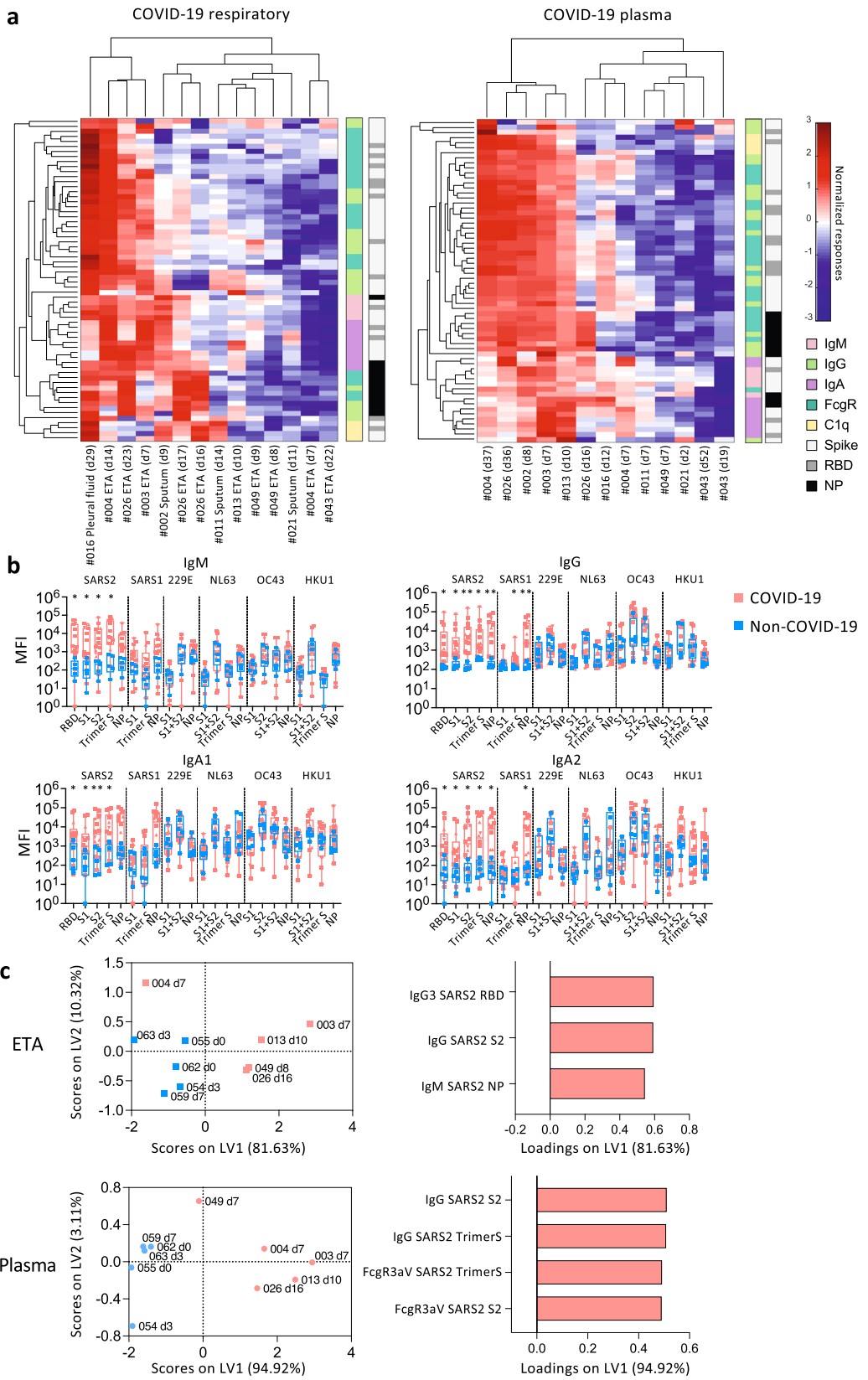

delayed or prolonged innate immune activation. Levels of sIL-6Rα were significantly higher in the severe/critical group than the mild/moderate group at both V1 ($p = 0.027$) and V7 ($p = 0.0302$), with the severe/critical group having higher sIL-6Rα and lower IL-6:sIL-6Rα ratios at V7 than V1 (Fig. 7a).

Anti-RBD IgG titres increased in both severity groups at discharge ($p = 0.0268$; $p = 0.0002$; Fig. 7b). The severe/critical group also displayed substantially higher microneutralization (MN) activity at discharge compared to admission ($p < 0.0001$). PLSDA revealed that at discharge the severe/critical group had

**Fig. 4 Higher SARS-CoV-2-specific IgM and IgG in COVID-19 ETA than non-COVID-19 ETA. a** Heatmaps with unsupervised clustering of SARS-CoV-2-specific antibodies in COVID-19 respiratory (endotracheal tube aspirate (ETA), sputum, or pleural fluid) and plasma samples. **b** median fluorescence intensity of IgM, IgG, IgA1, and IgA2 antibodies against receptor binding domain (RBD), spike proteins (S), and nucleoprotein (NP) of SARS-CoV-2 (SARS2), SARS-CoV-1 (SARS1), and other human coronaviruses (229E, NL63, OC43, HKU1) between COVID-19 and non-COVID-19 respiratory samples. The bounds of the box plot indicate the 25th and 75th percentiles, the bar indicates medians, and the whiskers indicate minima and maxima. Statistical significance was determined with a two-sided Mann-Whitney test. The P values for IgM against SARS2 RBD, SARS2 S1, SARS2 S2 and SARS2 Trimer S are 0.0103, 0.0143, 0.0143, 0.0103, respectively. The P values for IgG against SARS2 RBD, SARS2 S1, SARS2 S2, SARS2 Trimer S, SARS2 NP, SARS1 Trimer S and SARS1 NP are 0.0150, 0.0258, 0.0033, 0.0194, 0.0050, 0.0194, 0.0050, respectively. The P values for IgA1 against SARS2 RBD, SARS2 S1, SARS2 S2, and SARS2 Trimer S are 0.0437, 0.0258, 0.0072, 0.0258, respectively. The P values for IgA2 against SARS2 RBD, SARS2 S1, SARS2 S2, SARS2 Trimer S, SARS2 NP, SARS1 NP are 0.0258, 0.0258, 0.0103, 0.0339, 0.0143, 0.0258, respectively. **c** Partial Least-Squares Discriminant Analysis (PLSDA) scores and loading plots of ETA and plasma from five COVID-19 and five non-COVID-19 patients with the smallest difference in days post disease onset between ETA and plasma samples. $n_{COVID-19\ ETA} = 10$, $n_{COVID-19\ Sputum} = 3$, $n_{COVID-19\ pleural\ fluid} = 1$, $n_{Respiratory\ matched\ COVID-19\ plasma} = 13$, $n_{Non-COVID-19\ ETA} = 5$, $n_{Non-COVID-19\ sputum} = 1$. Source data are provided as a Source Data file.

higher IgM and IgG antibodies targeting SARS-CoV-2 proteins compared to the mild/moderate group (Fig. 7c).

COVID-19 patients in the severe/critical group had comparable frequencies of immune cells, while they had lower T cell and eosinophil frequencies ($p = 0.0011$; $p = 0.0473$) than the mild/moderate group at admission (V1; Fig. 7d). Interestingly, frequencies of mucosal-associated invariant T (MAIT) cells and γδ T cells negatively correlated with days in hospital ($p = 0.0022$ and $p = 0.0024$, respectively; Fig. 7d).

Overall, while cytokine levels were similar between the two severity groups, patients with more severe symptoms had more robust antibody responses towards the SARS-CoV-2.

**Dexamethasone did not alter immune responses in COVID-19 patients in blood.** Effects of dexamethasone, a corticosteroid anti-inflammatory drug, with/without remdesivir on immune responses in blood are unclear. We found very few differences in immune profiles between patients with/without dexamethasone. IL-8 and sIL-6Rα levels at discharge were significantly higher than at admission in the dexamethasone (with/without remdesivir) group, but similar levels were observed without treatment (Fig. 8a). Patients on treatment had lower anti-inflammatory IL-10 levels at discharge ($p = 0.0281$; Fig. 8a). Conversely, the humoral responses of patients receiving drugs were not compromised. Patients receiving dexamethasone (with/without remdesivir) generated robust SARS-CoV-2-specific antibody responses (Fig. 8b). Given that 29/33 severe/critical patients were on treatment, compared to 7/27 mild/moderate patients, high antibody levels in the drug group were likely due to disease severity rather than drug treatment. PLSDA revealed that patients prior to drug therapy had higher antibodies against the NP of human coronavirus OC43 rather than SARS-CoV-2, providing insights into potential drug treatment based on patient antibody responses at hospital admission (Fig. 8c). No significant differences were found in cellular responses, apart from lower T cell frequency in the drug group (Fig. 8d).

**Discussion**

Immunity to SARS-CoV-2 in the respiratory tract, the primary site of infection, is incompletely understood. We found differential inflammatory status in the respiratory tract and blood of COVID-19 patients, with high magnitude of MCP-1, IL-6, and IL-8 in respiratory specimens. While high SARS-CoV-2-specific IgG and IgM were detected in COVID-19 respiratory samples, IgG with FcγR-binding profiles were more prominent in blood. We found higher frequencies of neutrophils, intermediate CD14+CD16+monocytes, activated HLA-DR+CD38+ CD4+ T cells, EM-like CD4+ and CD8+ T cells in COVID-19 respiratory compared to blood samples. In blood, similar humoral

immune responses were observed in patients with/without dexamethasone treatment.

High levels of cytokines are commonly found in the blood of COVID-19 patients[29–32]. In respiratory samples, variable cytokine levels (IL-10, IL-17A, IL-18) were detected, while monocyte chemoattractants (MCP-1, MIP-1α, MIP-1β) and innate cytokines (IL-6, IL-10) were at high levels[18,19]. We found hypercytokinemia in respiratory samples compared to blood, especially IL-6, IL-8, and MCP-1, indicating an inflammatory environment that attracts leukocytes, including neutrophils and monocytes[33,34]. Since most patients did not have similar cytokine profiles in blood and respiratory samples after normalizing the cytokine levels within each sample type, measuring both blood and respiratory inflammation, especially in sputum, might be needed to accurately determine the inflammatory status of the patients. Interestingly, IFN-α2 level in ETA samples negatively correlated with days of hospital stay in our cohort. Similarly, IFN levels in nasopharyngeal samples of COVID-19 patients negatively correlated with viral load[35], indicating that viruses might be better controlled in the respiratory tract with higher IFN levels. Conversely, while higher plasma IL-12 levels were associated with more severe disease in COVID-19 patients[36], ETA IL-12 levels also negatively correlated with days in the hospital, potentially due to its enhancement of CD8+ T cell activation[37].

SARS-CoV-2-specific IgG and IgA were detected previously in BALF, sputum and saliva of COVID-19 patients[38–40]. We found detectable anti-RBD IgM, IgG and IgA in COVID-19 respiratory samples, with higher IgM and IgG than non-COVID-19 respiratory samples. Similar to plasma samples, neutralizing activities in respiratory specimens positively correlated with levels of anti-RBD IgM and IgG. While most attention is focused on IgA at mucosal surfaces, SARS-CoV-2-specific IgM was also detected in sputum, BALF and saliva from severely-ill COVID-19 patients[38,40]. Unlike IgA[41], anti-RBD IgM in the saliva of COVID-19 patients strongly correlated with serum levels[40]. Similar to its function in circulation, IgM can also mediate complement activation in the respiratory tract[41]. As IgM might affect immunopathology in the respiratory tract, this warrants further investigations.

Similar to previous studies, neutrophils dominated in COVID-19 respiratory samples[21]. Longitudinal ETA samples indicated increases in cellular infiltrates during disease progression, while the presence of CD16lo neutrophils showed recruitment of immature neutrophils likely derived from emergency myelopoiesis in bone marrow[42]. RNA-sequencing of COVID-19 BALF neutrophils found similar immature states[43]. Higher frequencies of intermediate CD14+CD16+ monocytes and activated CD38+HLA-DR+CD4+ T cells in respiratory samples revealed activated signatures in the respiratory tract. Although low in overall frequency, higher frequency of activated respiratory T cells was associated with improved survival in COVID-19[18].

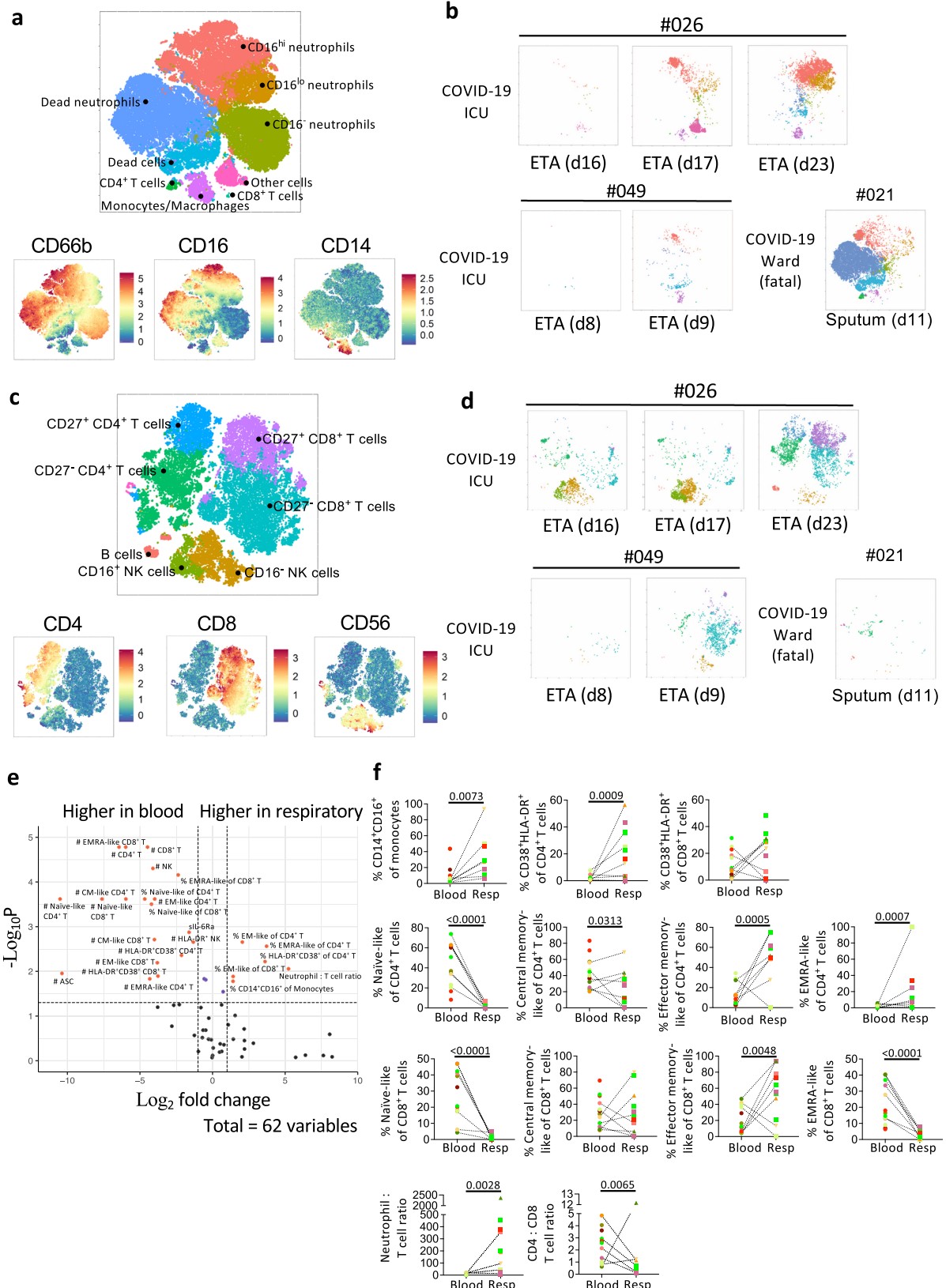

As an anti-inflammatory drug, dexamethasone can reduce proinflammatory cytokines including IL-6 and IL-8[44,45]. In blood, we found no significant difference in cytokine/chemokine levels between COVID-19 patients receiving dexamethasone and untreated patients. Although it has been speculated that dexamethasone can reduce the ability of B cells to produce antibodies[46], we showed similar antibody levels in patients with and without dexamethasone. Therefore, severely-ill COVID-19 patients might benefit from dexamethasone treatment as reported[5–7], and such treatment does not dampen humoral immunity.

There are limitations to the current study. Firstly, ETA samples were only collected from patients with severe disease requiring

**Fig. 5 Higher frequencies of activated immune cells and EM-like CD4⁺ and CD8⁺ T cells in COVID-19 respiratory compared to paired blood samples.** Flow Self-Organizing Map (FlowSOM) analyses of cellular content in the respiratory tract. **a–d** Metacluster of cells and expression level of markers in the **a** myeloid antibody panel and **c** lymphocyte antibody panel. Multiple endotracheal tube aspirate (ETA) samples from intensive care unit (ICU) patients #026 and #049 as well as one sputum sample from a fatal patient #021 were shown as example (**b**, **d**). **e** Volcano plot showing fold difference of 62 immunological features between paired respiratory and blood samples. **f** Comparisons of cellular immune features between respiratory and paired blood samples. $n_{Respiratory} = 14$, $n_{Respiratory\ matched\ blood} = 13$. Statistical significance was determined with a two-sided Mann-Whitney test. Dotted lines connect the most closely matched blood and respiratory samples from each patient. Colours indicate each patient. Source data are provided as a Source Data file.

invasive oxygen support, therefore, it is unclear whether COVID-19 patients with milder symptoms had less robust immune responses in the respiratory tract. Additionally, most patients in the severe/critical group received dexamethasone, which could be an intercorrelating factor for the differences observed between severity groups. Moreover, while the non-COVID-19 controls provided insights onto the immune status in hospitalized individuals, the comparisons would benefit from larger numbers of non-COVID patients with more homogenous diseases.

Overall, innate and adaptive immune responses are generated in respiratory and blood samples of COVID-19 patients. While immunological features detected in the peripheral blood might predict clinical outcomes, monitoring immune responses in the respiratory samples can be of benefit prior to initiation of therapeutic interventions for COVID-19 patients given the disparity observed between respiratory and blood specimens.

## Methods

**COVID-19 study participants and specimens**. We enrolled 60 SARS-CoV-2 PCR-positive patients admitted to Austin Health (Victoria, Australia) and 6 PCR-negative patients as negative controls for SARS-CoV-2 antibodies. We enrolled an additional 24 COVID-19 patients through Austin Hospital, Royal Melbourne Hospital, Alfred Hospital and Westmead Hospital. Two COVID-19 patients and three SARS-CoV-2 PCR-negative patients died during the study. Participants of the current study were not compensated. Peripheral blood was collected in heparinized, ethylenediaminetetraacetic acid (EDTA) or serum tubes during hospitalization and centrifuged to collect plasma or serum. Peripheral blood monocular cells (PBMCs) were isolated via Ficoll-Paque separation. Single-cell suspensions were isolated from tissues as previously described[47,48]. ETA samples were obtained as part of routine suctioning of the endotracheal tube airway and involved the passage of a catheter for suctioning into a sterile respiratory sample trap. Sputum samples were spontaneously collected into a sterile container. Respiratory samples from the Alfred Hospital were residual samples taken as part of routine care. Pleural fluid was collected by thoracentesis as part of a routine procedure. The thoracentesis involved the percutaneous insertion of a catheter into the pleural space and collection of pleural fluid into a sterile container. Demographic, clinical and sampling information for COVID-19 patients are described in Supplementary Table 1. For the 6 respiratory samples with undetectable cytokine/chemokine levels, we were still able to detect antibody levels, reflecting the high quality and integrity of the samples (Supplementary Table 6).

**Ethics statement**. Experiments conformed to the Declaration of Helsinki Principles and the Australian National Health and Medical Research Council Code of Practice. Written informed consent was obtained from all donors prior to the study. The study was approved by the Austin Health (HREC/63201/Austin-2020), the Alfred Hospital Ethics Committee (Project 182/20), Western Sydney Local Area Health District (WSLHD) Human Research Ethics Committee (HREC) (2020/ETH00989), Melbourne Health (HREC/66341/MH-2020 and HREC/17/MH/53) and the University of Melbourne (#2057366.1, #2056901.1 and #1955465.3) Human Research Ethics Committees.

**Genomic sequencing and bioinformatic analysis**. Extracted RNA from RT-PCR positive samples underwent tiled amplicon PCR and Illumina short-read sequencing, quality control, consensus sequence generation and alignment as previously described[49]. A single sequence per patient was used for phylogenetic analysis[22], with a maximum-likelihood phylogenetic tree generated using IQ-Tree (v2.1.-, options "-mset GTR + G4 -bb 1000")[50] and visualized using the *ggtree* package (v.1.14.6) in R (v3.5.3)[51]. Genomic clusters were defined using a hierarchical clustering algorithm; genomic transmission networks grouped multiple clusters supported by epidemiological and genomic data.

**Phenotypic whole blood immune analyses**. Fresh whole blood (200 µl per stain) was used to measure CD4⁺CXCR5⁺ICOS⁺PD1⁺ follicular T cells (Tfh) and

CD3⁻CD19⁺CD27^hiCD38^hi antibody-secreting B cell (ASC; plasmablast) populations as described[15,52] as well as activated HLA-DR⁺CD38⁺CD8⁺ and HLA-DR⁺CD38⁺CD4⁺ T cells, intermediate CD14⁺CD16⁺ and classical CD14⁺ monocytes, activated CD3⁻CD56⁺ NK cells, MAIT cells, γδ-T cells, as per the specific antibody panels (Supplementary Table 7; gating strategy is presented in Supplementary Fig. 3b, c). After whole blood was stained for 20 min at room temperature in the dark, samples were lysed with BD FACS Lysing solution (BD Biosciences, San Jose, California, USA), washed and fixed with 1% PFA. AccuCheck Counting Beads (Thermo Fisher Scientific, Carlsbad, CA, USA) were added for calculating absolute numbers just prior to acquisition. All samples were acquired on a LSRII Fortessa (BD) using the software BD FACS DIVA v8.0.1. Flow cytometry data were analyzed using FlowJo v10 software.

**Phenotypic immune analyses in respiratory samples**. Respiratory samples (ETA, sputum, BAL, or pleural fluid) were diluted in PBS and were filtered through a 45 µm filter prior to the separation of respiratory fluid and cellular contents by centrifugation. The respiratory fluid was frozen at −20 °C, and the cell pellet (ETA, sputum, or pleural fluid) was washed with EDTA-BSS. Washed cells were stained with FcR block (Miltenyi Biotec, Bergisch Galdbach, Germany) for 15 min followed by 30 min staining on ice with specific antibody panels (Supplementary Table 7). After fixing with 1% PFA, the samples were acquired on a LSRII Fortessa (BD) using the software BD FACS DIVA v8.0.1. AccuCheck Counting Beads (Thermo Fisher Scientific) were added for calculating absolute numbers just prior to acquisition. Flow cytometry data were analyzed using FlowJo v10 software.

**SARS-CoV-2 RBD ELISA**. RBD-specific ELISA for detection of IgM, IgG and IgA antibodies was performed as previously described[31,53,54], using flat bottom Nunc MaxiSorp 96-well plates (Thermo Fisher Scientific) for antigen coating (2 µg/ml), blocking with PBS with w/v 1% BSA and serial dilutions in PBS with v/v 0.05% Tween and w/v 0.5% BSA. Plates were read on a Multiskan plate reader (Labsystems, Vantaa, Finland) using the Thermo Ascent Software for Multiskan v2.4. Inter- and intra-experimental measurements were normalised using positive control plasma from a COVID-19 patient run on each plate. Endpoint titres were determined by interpolation from a sigmoidal curve fit (all R-squared values >0.95; GraphPad Prism 9) as the reciprocal dilution of plasma that produced >15% (for IgA and IgG) or >30% (for IgM) absorbance of the positive control at a 1:31.6 (IgG and IgM) or 1:10 dilution (IgA). Seroconversion was defined when titres were above the mean titre (plus 2 standard deviations) of non-COVID-19 control respiratory or plasma samples.

**Microneutralization assay**. Microneutralization activity of serum samples was assessed as previously described[55]. SARS-CoV-2 isolate CoV/Australia/VIC01/2020[56] was propagated in Vero cells (ATCC #CCL-81) and stored at −80 °C. Sera were heat-inactivated at 56 °C for 30 min and serially diluted. Residual virus infectivity in the serum/virus mixtures was assessed in quadruplicate wells of Vero cells incubated in serum-free media containing 1 µg/ml of TPCK trypsin at 37 °C and 5% CO₂. The viral cytopathic effect was read on day 5. The neutralizing antibody titer was calculated using the Reed-Muench method[55].

**SARS-CoV-2 surrogate virus neutralisation test (sVNT)**. The plasma samples were tested in neat, and the respiratory samples were tested at 1:9 dilution or at their original dilutions for more diluted samples. The sVNT blocking ELISA assay (manufactured by GenScript, NJ, USA) was carried out essentially as described[54], which detects circulating neutralizing SARS-CoV-2 antibodies that block the interaction between RBD and ACE2 on the cell surface receptor of the host. A HRP-conjugated recombinant SARS-CoV-2 RBD fragment bound to any circulating neutralizing anti-RBD antibodies preventing capture by the human ACE2 protein in the well, which was subsequently removed in the following wash step. Substrate reaction incubation time was 20 min at room temperature and results were read spectrophotometrically. Colour intensity was inversely dependent on the titre of anti-SARS-CoV-2 neutralizing antibodies.

**Coupling of carboxylated beads**. As previously described[25], a custom multiplex bead array was designed and coupled with SARS-CoV-2 spike 1 (Sino Biological), spike 2 (ACRO Biosystems), RBD (BEI Resources) and nucleoprotein (ACRO Biosystems), as well as SARS and hCoV (229E, NL63, HKU1, OC43) spikes and

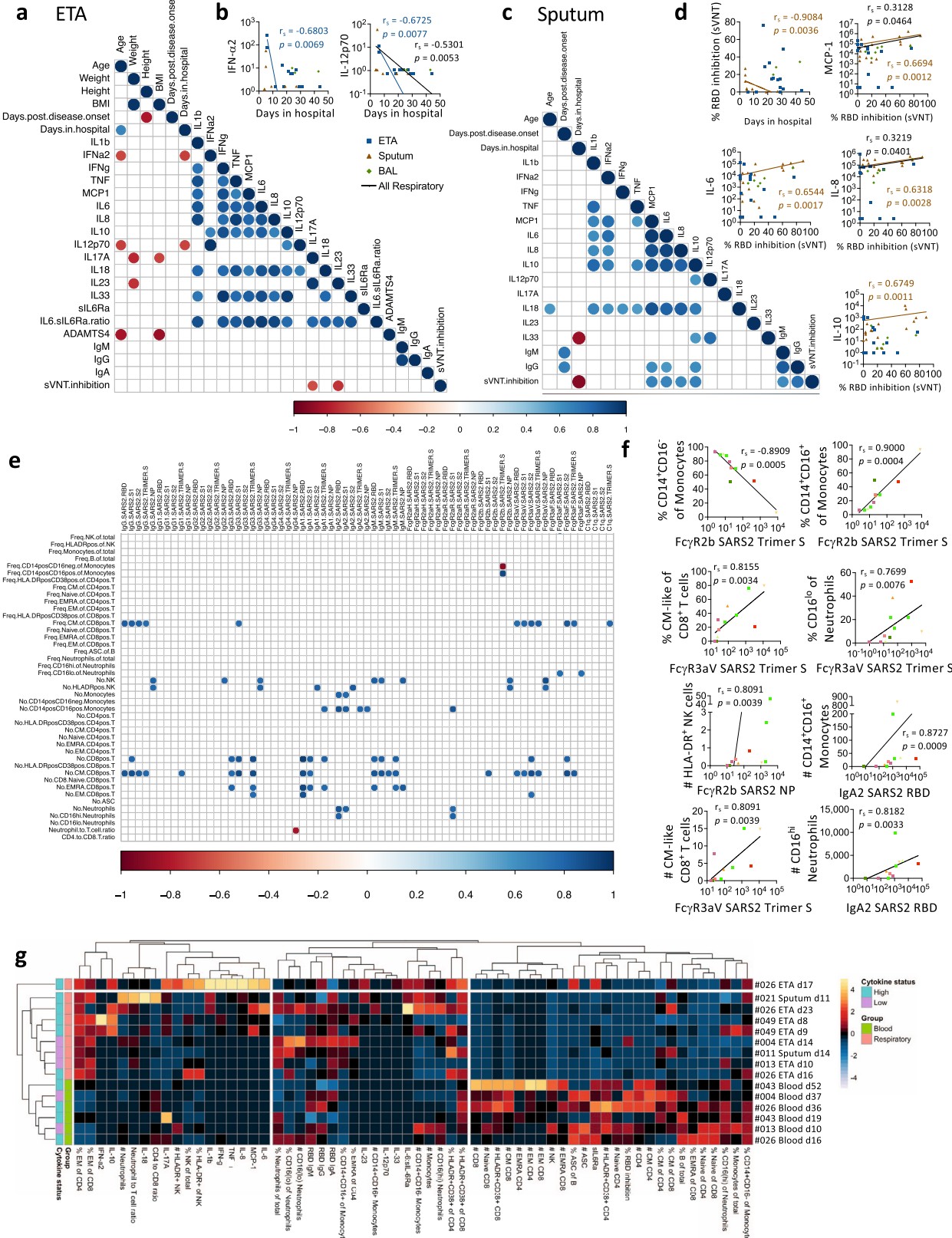

**Fig. 6 Levels of ETA IFN-a2 and IL-12p70 and sputum neutralizing activity negatively correlate with days of hospital stay. a–d** Correlation matrix and correlation graphs between immune features in COVID-19 respiratory (endotracheal tube aspirate (ETA) or sputum) samples and clinical features. $n_{ETA} = 15$, $n_{Sputum} = 20$, $n_{BAL} = 6$. **e–f** Correlation matrix and graphs between multiplex and non-multiplex immune features in COVID-19 respiratory samples. $n_{COVID-19\ ETA} = 10$, $n_{COVID-19\ Sputum} = 3$, $n_{COVID-19\ pleural\ fluid} = 1$. Correlation was determined with a two-tailed Spearman's correlation and $p$ values of the correlation matrix were adjusted with False Discovery Rate adjustment. **g** Heatmaps with unsupervised clustering of serological and cellular features in COVID-19 respiratory and blood samples. Source data are provided as a Source Data file.

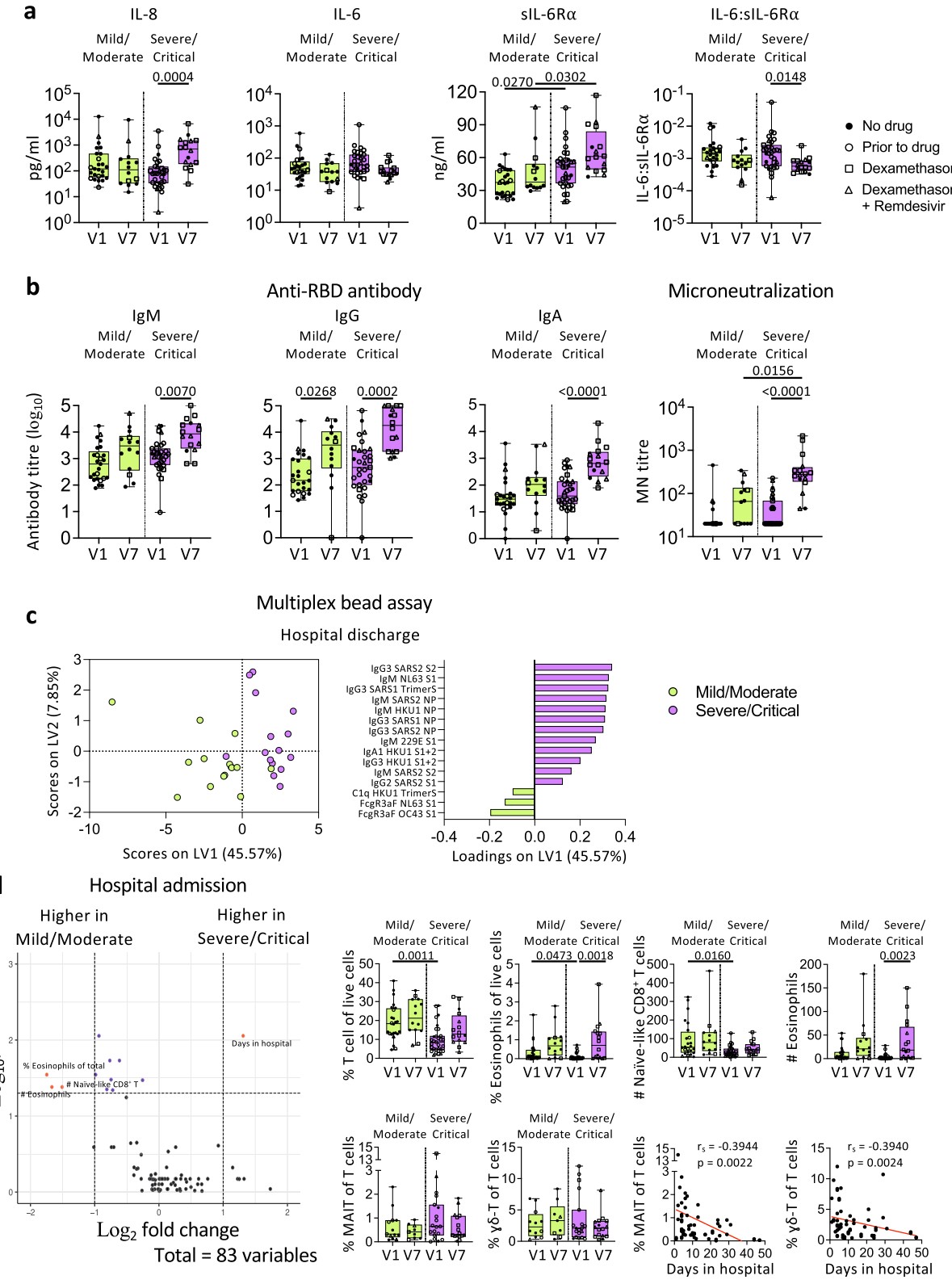

nucleoproteins (Sino Biological) (Supplementary Table 5). In addition, SARS-CoV-2 and HKU-1 spike trimers (kind gifts from Adam K. Wheatley), as well as SARS-CoV and NL63 spike trimers (BPS Bioscience) were also coupled. Tetanus toxoid (Sigma-Aldrich), influenza hemagglutinin (H1Cal2009; Sino Biological) and SIV gp120 (Sino Biological) were included in the assay as positive and negative controls respectively. Antigens were covalently coupled to magnetic carboxylated beads (Bio

Rad) using a two-step carbodiimide reaction and blocked with 0.1% BSA, before being resuspended and stored in PBS 0.05% sodium azide.

**Luminex bead-based multiplex assay**. Using the coupled beads mentioned above, a custom CoV multiplex assay was formed to investigate the isotypes and

**Fig. 7 COVID-19 patients with higher NIH scores display more robust humoral immune responses towards SARS-CoV-2 in blood. a** Levels of cytokines, soluble IL-6 receptor α (sIL-6Rα), and IL-6:sIL-6Rα ratio, **b** anti-RBD IgM, IgG, and IgA titres, microneutralization titres and **c** Partial Least-Squares Discriminant Analysis (PLSDA) scores and loadings plot of antibodies against human coronavirus between mild/moderate and severe/critical COVID-19 patients. **d** Volcano plot showing fold difference of 83 immunological features in blood samples between mild/moderate and severe/critical COVID-19 patients, and comparisons of cellular subset frequencies and correlation with days stayed in hospital. $n_{Mild\text{-}Moderate\ V1} = 25$, $n_{Mild\text{-}Moderate\ V7} = 14$, $n_{Severe\text{-}Critical\ V1} = 31$, $n_{Severe\text{-}Critical\ V7} = 16$. The bounds of the box plot indicate the 25th and 75th percentiles, the bar indicates medians, and the whiskers indicate minima and maxima. Statistical significance was determined with a two-sided Kruskal-Wallis test followed by Dunn's multiple comparisons test. Partial Least-Squares Discriminant Analysis was performed for antibodies measured with multiplex bead array assay. Volcano plots were created using a two-sided Wilcoxon rank-sum test and statistics were corrected with FDR adjustment. Correlation was determined with Spearman's correlation. V1, hospital admission; V7, hospital discharge. Source data are provided as a Source Data file.

subclasses of pathogen-specific antibodies present in collected plasma samples[25]. Briefly, 20 µl of working bead mixture (1000 beads per bead region) and 20 µl of diluted plasma (final dilution 1:200) or 20 µl of diluted respiratory secretions (final dilution 1:800) were added per well and incubated overnight at 4 °C on a shaker. Fourteen different detectors were used to assess pathogen-specific antibodies. Single-step detection was done using phycoerythrin (PE)-conjugated mouse anti-human pan-IgG, IgG1-4 and IgA1-2 (Southern Biotech; 1.3 µg/ml, 25 µl/well). C1q protein (MP Biomedicals) was first biotinylated (Thermo Fisher Scientific), then tetramerized with Streptavidin R-PE (SAPE; Thermo Fisher Scientific) before dimers or tetrameric C1q-PE were used for single-step detection. For the detection of FcγR-binding, soluble recombinant FcγR dimers (higher affinity polymorphisms FcγRIIa-H131, lower affinity polymorphisms FcγRIIa-R131, FcγRIIb, higher affinity polymorphisms FcγRIIIa-V158 and lower affinity polymorphisms FcγRIIIa-F158; 1.3 µg/ml, 25 µl/well; kind gifts from Bruce D. Wines and P. Mark Hogarth) were first added to the beads, washed, and followed by the addition of SAPE. For the detection of IgM, biotinylated mouse anti-human IgM (mab MT22; Mabtech; 1.3 µg/ml, 25 µl/well) was first added to beads, washed, followed by SAPE. Assays were read on the Flexmap 3D (Luminex) using the Luminex xPONENT v4.3 and performed in duplicates.

**Data normalization**. For all multivariate analysis, Tetanus, H1Cal2009, and BSA antigens (positive controls) were removed, as well as SIV (negative control). Low signal features were removed when the 75th percentile response for the feature was lower than the 75th percentile of the BSA positive control. Right shifting was performed on each feature (detector–antigen pair) individually if it contained any negative values, by adding the minimum value for that feature back to all samples within that feature. Following this, all data were log-transformed using the following equation, where x is the right-shifted data and y is the right-shifted log-transformed data: $y = \log10(x + 1)$. This process transformed the majority of the features to having a normal distribution. In all the subsequent multivariate analyses, the data were further normalized by mean centering and variance scaling each feature using the z-score function in Matlab 2017b (MathWorks, Natick, MA). Plasma and respiratory samples were analysed separately. When analysing samples at time of hospital discharge, to adjust for the confounder of time from symptom onset, each of the features were iteratively regressed with ordinary least squares regression, using the residuals as input for the analysis[57].

**Feature selection using elastic Net/PLSDA**. To determine the minimal set of features (signatures) needed to predict categorical outcomes (COVID-19 diagnosis, NIH scores, drug therapies), a three-step process was developed[58]. First, the data were randomly sampled without replacement to generate 2000 subsets. The resampled subsets spanned 80% of the original sample size, or sampled all classes at the size of the smallest class for categorical outcomes, which corrected for any potential effects of class size imbalances during regularization. Elastic-Net regularization was then applied to each of the 2000 resampled subsets to reduce and select features most associated with the outcome variables. The Elastic-Net hyperparameter, α, was set to have equal weights between the L1 norm and L2 norm associated with the penalty function for least absolute shrinkage and selection (LASSO) and ridge regression, respectively[59]. By using both penalties, Elastic-Net provides sparsity and promotes group selection. The frequency at which each feature was selected across the 2000 iterations was used to determine the signatures by using a sequential step-forward algorithm that iteratively added a single feature into the PLSDA model starting with the feature that had the highest frequency of selection, to the lowest frequency of selection. Model prediction performance was assessed at each step and evaluated by 10-fold cross-validation classification error for categorical outcomes. The model with the lowest classification error within a 0.01 difference between the minimum classification error was selected as the minimum signature. If multiple models fell within this range, the one with the least number of features was selected and if there was a large disparity between calibration and cross-validation error (over-fitting), the model with the least disparity and best performance was selected.

**PLSDA**. Partial least squares discriminant analysis (PLSDA), performed in Eigenvectors PLS toolbox 8.2 in Matlab 2017b, was used in conjunction with Elastic-Net, described above, to identify and visualize signatures that distinguish categorical outcomes (COVID-19 diagnosis, NIH scores, drug therapies). This supervised method assigns a loading to each feature within a given signature and identifies the linear combination of loadings (a latent variable, LV) that best separates the categorical groups. A feature with a high loading magnitude indicates greater importance for separating the groups from one another. Each sample is then scored and plotted using their individual response measurements expressed through the LVs. The scores and loadings can then be cross-referenced to determine which features are loaded in association with which categorical groups (positively loaded features are higher in positively scoring groups, etc.). All models go through 10-fold cross-validation, where iteratively 10% of the data is left out as the test set, and the rest is used to train the model. Model performance is measured through calibration error (average error in the training set) as well as cross-validation error (average error in the test set), with values near 0 being best. All models were orthogonalized to enable clear visualization of results. PLSDA scores and loadings plots were plotted in Prism v8.

**Hierarchical clustering**. We visualized the clustering of DRASTIC respiratory and blood samples based on only SARS-CoV-2 antigens or all features using unsupervised average linkage hierarchical clustering of normalized data using MATLAB 2017b. Euclidean distance was used as the distance metric.

**Cytokine analysis**. Patients' plasma and respiratory samples were measured for IL-1β, IFN-α2, IFN-γ, TNF, MCP-1 (CCL2), IL-6, IL-8 (CXCL8), IL-10, IL-12p70, IL-17A, IL-18, IL-23 and IL-33 using the LEGENDplex™ Human Inflammation Panel 1 kit, as per manufacturer's instructions (BioLegend, San Diego, CA, USA). Samples were acquired on a BD CantoII using the software BD FACS DIVA v8.0.1. Data were analyzed using LEGENDplex™ Data Analysis Software v7.1.

**sIL-6Rα and ADAMTS4 ELISAs**. Soluble protein levels were all measured using DuoSet ELISA kits for each protein (R&D Systems, Minneapolis, MN, USA) according to the manufacturer's instructions. DuoSet ELISA ancillary reagent kit (R&D Systems) was used for respiratory fluids and in-house reagents with the same composition were used for plasma samples. In brief, 96-well R&D ELISA micro-plates (respiratory fluids) or 96-well Nunc Maxisorp ELISA plates (ThermoFisher, plasma) were coated with capture antibody overnight, followed by blocking with 1% w/v BSA for a minimum of 1 h. Samples and standard proteins were added and incubated for 2 h at room temperature, followed by detection antibody for a further 2 h. Lastly, streptavidin-HRP, substrate solution and stop solution (2 N $H_2SO_4$) were added subsequently for 20 min each. Plates were read on a Multiskan plate reader (Labsystems) using the Thermo Ascent Software for Multiskan v2.4. Plasma samples were diluted in 1:300 for sIL-6Rα ELISAs. Respiratory fluids were diluted in 1:50/1:150 for sIL-6Rα ELISA accounting in the original dilution factors and tested without further dilution for ADAMTS4 ELISA.

**Computational flow cytometry analysis**. Computational analysis of data was performed using the Spectre R package (v0.4.1)[26] (https://github.com/ImmuneDynamics/Spectre). Samples were initially prepared in FlowJo, and populations of interest were exported as CSV files containing raw (scale value) data. In R (v4.0.2), data were subject to arcsinh transformation and clustering using FlowSOM (v1.20.0)[27]. For visualisation, cells in the myeloid panel were subjected to sample-weighted downsampling based on absolute cells/uL counts in the blood, whereas cells in the lymphoid panel were downsampled to preserve samples with low cell numbers. Cells from the lymphoid panel, and downsampled cells from the myeloid panel, were then distributed in 2D via dimensionality reduction (DR) using Fast Interpolation-based t-distributed Stochastic Neighbour Embedding (FIt-SNE, v1.2.1)[28]. Data from both the lymphoid and myeloid panel were subject to two rounds of clustering and DR. The initial round of clustering and DR was used to filter out cellular debris and non-immune cells exhibiting high autofluorescence, using arcsinh transformed expression of CD45, CD3, CD4, CD8, CD14, CD16, CD19, CD64, CD66b, CD38, HLA-DR, and Live-Dead for the myeloid

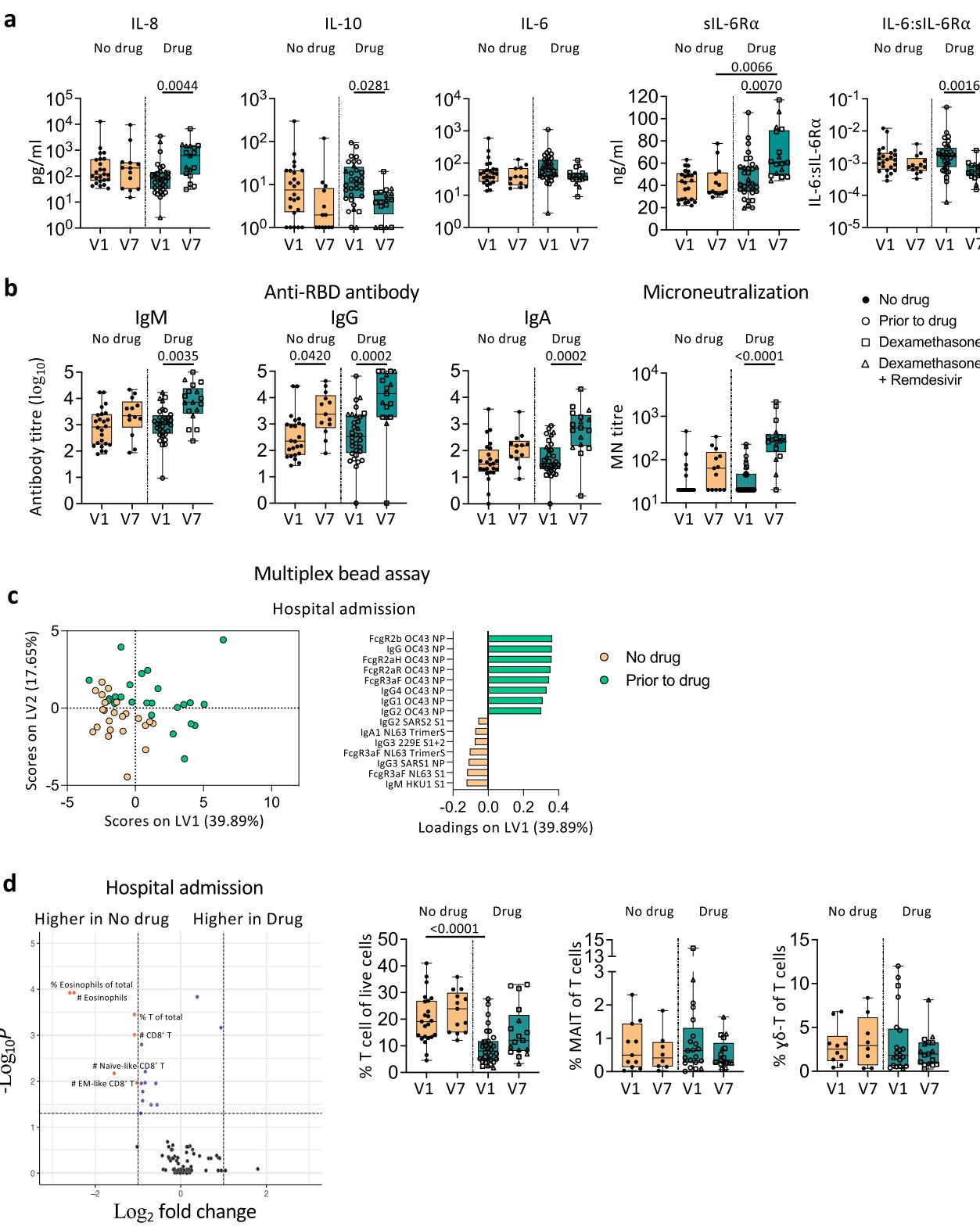

panel; and CD45, CD3, CD4, CD8, TCR-γδ, CD45RA, CD27, CD56, CD19, CD16, CD14, CD38, HLA-DR, PD-1, and Live-Dead for the lymphoid panel. A second round of clustering and DR was then used for detailed immunophenotyping of cells in the respiratory tract, using arcsinh transformed expression of CD4, CD8, CD14, CD16, CD38, CD45, CD64, CD66b, HLA-DR, and Live-Dead for the myeloid panel; and CD45, CD3, CD4, CD8, TCR-γδ, CD45RA, CD27, CD56, CD19, CD16, CD38, and HLA-DR for the lymphoid panel.

Immune cell lineages were manually annotated based on marker expression: neutrophils (FSC$^{int}$SSC$^{int}$CD66b$^+$), monocytes (FSC$^{int}$CD14$^+$), B cells (CD19$^+$), NK cells (CD56$^+$), gamma-delta T cells (TCR-γδ$^+$), CD4$^+$ T cells (CD4$^+$CD14$^-$), CD8$^+$ T cells (CD8$^+$), and dead cells (Live-Dead$^+$). Interestingly, CD3 expression on T cell subsets was not apparent in respiratory samples, and we did not find distinct eosinophil phenotypes. Subsequently, population subsets were manually annotated based on marker expression: neutrophils (CD16$^{hi}$, CD16$^{lo}$, CD16$^-$, dead neutrophils

**Fig. 8 Dexamethasone treatment does not alter humoral immune responses towards SARS-CoV-2 in COVID-19 patients in blood. a** Levels of cytokines, soluble IL-6 receptor α (sIL-6Rα), and IL-6:sIL-6Rα ratio; **b** anti-RBD IgM, IgG, and IgA titres, microneutralization titres; **c** Partial Least-Squares Discriminant Analysis (PLSDA) scores and loadings plot of antibodies against human coronaviruses; **d** cellular immune subset frequencies between COVID-19 patients with or without dexamethasone treatment (with/without remdesivir). $n_{No\ drug\ V1} = 24$, $n_{No\ drug\ V7} = 13$, $n_{Drug\ V1} = 32$, $n_{Drug\ V7} = 17$. The bounds of the box plot indicate the 25th and 75th percentiles, the bar indicates medians, and the whiskers indicate minima and maxima. Statistical significance was determined with a two-sided Kruskal-Wallis test followed by Dunn's multiple comparisons test. Partial Least-Squares Discriminant Analysis was performed for antibodies measured with multiplex bead array assay. Volcano plots were created using a two-sided Wilcoxon rank-sum test and statistics were corrected with FDR adjustment. V1, hospital admission; V7, hospital discharge. Source data are provided as a Source Data file.

Live-Dead[+]), monocytes (classical CD14[+]CD16[−] and intermediate CD14[+]CD16[+]), B cells (naïve CD27[−]CD38[−], memory CD27[+]CD38[−], ASC CD27[+]CD38[+]), NK cells (CD56[bri] and CD56[dim]), CD4[+] T cells (naïve-like CD45RA[+]CD27[+], EMRA-like CD45RA[+]CD27[−], CM-like CD45RA[−]CD27[+], and EM-like CD45RA[−]CD27[−]), and CD8[+] T cells (naïve-like CD45RA[+]CD27[+], EMRA-like CD45RA[+]CD27[−], CM-like CD45RA[−]CD27[+], and EM-like CD45RA[−]CD27[−]). Given that NK cells were classified as CD56[+] cells, the subset might include other unconventional T cells. Subsets were evaluated for expression of CD38, HLA-DR, and PD-1 expression using manual gating in FlowJo.

Volcano plots and heatmaps were created using the Spectre R package[26], where comparisons were performed using a Wilcoxon rank-sum test (equivalent to the Mann-Whitney test) with the wilcox.test function in R. Statistics displayed in volcano plots were corrected with a False Discovery Rate (FDR) adjustment.

**Statistical analyses**. Statistical significance was assessed using two-sided Mann-Whitney, Wilcoxon signed-rank test or Kruskal-Wallis test with Dunn's correction for multiple comparisons in Prism 9 (GraphPad) unless stated otherwise. Correlations were assessed using two-tailed Spearman's correlation coefficient ($r_s$) and visualized in R v3.6.2 as heatmaps using the corrplot package or using the online Morpheus heatmap software (https://software.broadinstitute.org/morpheus; the Broad Institute, MA, USA) and p-values of correlations were corrected for multiple comparisons by FDR in R v3.6.2. $P$-values lower than 0.05 were considered statistically significant.

**Reporting summary**. Further information on research design is available in the Nature Research Reporting Summary linked to this article.

## Data availability

All data generated or analysed during this study are included in this published article (and its supplementary information files). A Source Data file is provided with this paper. All relevant data are also available from the authors. The viral sequences isolated from nasal swabs that support the findings of this study are available on the GISAID database with ID numbers provided in the Source data file. Registration to access the database is free and open to anyone using the link: https://www.gisaid.org/registration/register/.

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

## Acknowledgements

We acknowledge all DRASTIC (The use of cytokines as a preDictoR of disease Severity in criTically Ill COVID-19) investigators from Austin Health, and thank the participants involved. This project obtained samples from the COVID-19 Biobank based at the Alfred Hospital and acknowledge all its investigators and the participants who contributed samples. We acknowledge Janine Roney's support for this work and funding from the Lord Mayor's Charitable Foundation and the Ray William Houston Trust. This research included samples and data from the Sentinel Travelers Research Preparedness Platform for Emerging Infectious Diseases (SETREP-ID). We acknowledge all SETREP-ID investigators and sites, and thank all participants involved. The authors thank Barbara Scher for setting up the ethics and governance for the SETREP-ID platform and the Australian Partnership for Preparedness Research for Infectious Disease Emergencies (APPRISE) for ongoing funding of SETREP-ID, Ajantha Rhodes, Judy Chang, Ashanti Dantanarayana and Rosalyn Cao who contributed to the SETREP-ID biobank. SETREP-ID is supported by funding through the National Health and Medical Research Council Centre of Research Excellence (NHMRC CRE), the Australian Partnership for Preparedness Research on Infectious Disease Emergencies (APPRISE AppID 1116530), the Snow Medical Foundation, the Jack Ma Foundation and the A2 Milk Company. We thank Austin Health Foundation for support of the study. We acknowledge Francesca L. Mordant and Kanta Subbarao for their contributions to the microneutralization assay. We acknowledge Adam K. Wheatley for kindly providing SARS-CoV-2 and HKU-1 spike trimers and Bruce D. Wines and P. Mark Hogarth for kindly providing FcγR dimers. The RBD proteins were produced under HHSN272201400008C and obtained through BEI Resources, NIAID, NIH: Spike Glycoprotein RBD from SARS-Related Coronavirus 2, Wuhan-Hu-1 with C-Terminal Histidine Tag, Recombinant from HEK293F Cells, NR-52366. We thank the staff at the diagnostic microbiology laboratories at Austin Pathology, Melbourne Pathology, Dorevitch Pathology, 4Cyte Pathology, Microbiological Diagnostic Unit Public Health Laboratory, Northern Pathology Victoria, Eastern Health Pathology for performing initial diagnostic testing for the detection of SARS-CoV-2 nucleic acid. We acknowledge Bridie Clemens for reviewing our manuscript. This work was supported by the NHMRC Leadership Investigator Grant to KK (#1173871), Research Grants Council of the Hong Kong Special Administrative Region, China (#T11-712/19-N) to KK, the MRFF Award (#1202445) to KK and AWC, NIH contract CIVC-HRP (HHS-NIH-NIAID-BAA2018) to PGT and KK, NHMRC Senior Principal Research Fellowship (#1117766) to DIG, NHMRC Emerging Leadership Level 1 Investigator Grant to THON (#1194036) and NHMRC Early Career Fellowships to HFK (#1160333), CLG (#1160963) and JAT (#1139902). PGT is supported by NIH NIAID R01 AI136514-03 and ALSAC at St. Jude. WZ and JRH are supported by the Melbourne Research Scholarship from The University of Melbourne. LH is supported by the Melbourne International Research Scholarship (MIRS) and the Melbourne International Fee Remission Scholarship (MIFRS) from The University of Melbourne. We acknowledge the Melbourne Cytometry Platform (Peter Doherty Institute and Melbourne Brain Centre nodes) for provision of flow cytometry services.

## Author contributions

K.K., T.H.O.N., C.L.G. and J.A.T. supervised the study. K.K., T.H.O.N., C.L.G., J.A.T., W.Z., B.Y.C., K.J.S., L.K., H.F.K., S.N. and A.W.C. designed the experiments. W.Z., B.Y.C., K.J.S., L.K., E.R.H., L.H., L.C.R., J.R.H., L.F.A., H.F.K., J.A.N., M.J.G., S.N. and T.H.O.N. performed and analysed experiments. W.Z., T.M.A., S.K.S.B., C.Y.L., P.A., T.S., N.L.S. and K.B.A. analysed data. D.F.B., F.A., F.K. and P.G.T. provided reagents. F.J., E.M., J.K., K.Y.L.C., G.D., A.C., J.E.D., N.E.H., O.C.S., J.A.T., C.L.G., S.F.K., L.B., J.W., J.H.M., E.E.V., A.L.C., J.A., I.T. recruited the patient cohorts and provided clinical data. W.Z., S.L.L., L.M.W., N.J.C.K., D.I.G., L.K.M., P.G.T., S.N., K.B.A., A.W.C., J.A.T., C.L.G., T.H.O.N. and K.K. provided intellectual input into the study design and data interpretation. W.Z., T.H.O.N. and K.K. wrote the manuscript. All authors reviewed and approved the manuscript.

## Competing interests

The Icahn School of Medicine at Mount Sinai has filed patent applications relating to SARS-CoV-2 serological assays (U.S. Provisional Application Numbers: 62/994,252, 63/018,457, 63/020,503 and 63/024,436) and NDV-based SARS-CoV-2 vaccines (U.S. Provisional Application Number: 63/251,020) which list Florian Krammer as co-inventor. Fatima Amanat is also listed on the serological assay patent application as co-inventors. Patent applications were submitted by the Icahn School of Medicine at Mount Sinai. Mount Sinai has spun out a company, Kantaro, to market serological tests for SARS-CoV-2. Florian Krammer has consulted for Merck and Pfizer (before 2020), and is currently consulting for Pfizer, Third Rock Ventures, Seqirus and Avimex. The Krammer laboratory is also collaborating with Pfizer on animal models of SARS-CoV-2. Dale Godfrey is listed as co-inventor on patent applications relating to SARS-CoV-2 vaccines and lateral flow assay for neutralizing antibodies. Paul Thomas is on the SAB of Immunoscape and Cytoagents and has consulted for JNJ. Paul Thomas has received travel support and/or honoraria from Illumina and 10X Genomics and has patents related to TCR discovery and expression. All authors declare no other competing interests.

## Additional information

[1]Department of Microbiology and Immunology, University of Melbourne, at the Peter Doherty Institute for Infection and Immunity, Melbourne, Victoria 3000, Australia. [2]Global Station for Zoonosis Control, Global Institution for Collaborative Research and Education (GI-CoRE), Hokkaido University, Sapporo, Japan. [3]Faculty of Veterinary and Agricultural Sciences, University of Melbourne, Melbourne, Victoria 3000, Australia. [4]Sydney Cytometry Core Research Facility, Charles Perkins Centre, Centenary Institute and University of Sydney, Sydney, NSW, Australia. [5]Sydney Institute for Infectious Diseases, University of Sydney, Sydney, NSW, Australia. [6]Department of Biomedical Engineering, University of Michigan, Michigan, USA. [7]Department of Immunology, St Jude Children's Research Hospital, Memphis, TN, USA. [8]Department of Infectious Diseases, Austin Health, Heidelberg, VIC, Australia. [9]Department of Radiology, Austin Health, Heidelberg, VIC, Australia. [10]Microbiological Diagnostic Unit Public Health Laboratory, Department of Microbiology & Immunology, University of Melbourne at the Peter Doherty Institute for Infection and Immunity, Melbourne, Australia. [11]Department of Infectious Diseases, Monash University and Alfred Hospital, Melbourne, VIC, Australia. [12]Department of Infectious Diseases, Monash Medical Centre, Melbourne, VIC, Australia. [13]Centre for Virus Research, The Westmead Institute for Medical Research, Westmead, NSW, Australia. [14]School of Medical Sciences, Faculty of Medicine and Health, University of Sydney, Sydney, NSW, Australia. [15]Sydney Infectious Diseases, Faculty of Medicine and Health, University of Sydney, Westmead, NSW, Australia. [16]Department of Infectious Diseases, University of Melbourne, at the Peter Doherty Institute for Infection and Immunity, Melbourne, VIC 3000, Australia. [17]Victorian Infectious Diseases Services, The Royal Melbourne Hospital and Doherty Department University of Melbourne, at the Peter Doherty Institute for Infection and Immunity, Melbourne 3000 VIC, Australia. [18]Department of Microbiology, Icahn School of Medicine at Mount Sinai, New York, NY, USA. [19]Graduate School of Biomedical Sciences, Icahn School of Medicine at Mount Sinai, New York, NY, USA. [20]Charles Perkins Centre, University of Sydney, Sydney, NSW, Australia. [21]Viral Immunopathology Laboratory, Discipline of Pathology, School of Medical Sciences, University of Sydney, Sydney, NSW, Australia. [22]Sydney Nano, University of Sydney, Sydney, NSW 2006, Australia. [23]Victorian Infectious Diseases Reference Laboratory, The Royal Melbourne Hospital at The Peter Doherty Institute for Infection and Immunity, Melbourne, VIC, Australia. [24]Department of Critical Care, University of Melbourne, Parkville, VIC, Australia. [25]Data Analytics Research and Evaluation (DARE) Centre, Austin Health and University of Melbourne, Heidelberg, VIC, Australia. [26]Centre for Antibiotic Allergy and Research, Department of Infectious Diseases, Austin Health, Heidelberg, VIC, Australia. [27]Department of Infectious Diseases, Peter McCallum Cancer Centre, Melbourne, VIC, Australia. [28]National Centre for Infections in Cancer, Peter McCallum Cancer Centre, Melbourne, VIC, Australia. [29]Department of Medicine (Austin Health), University of Melbourne, Heidelberg, VIC, Australia. [30]These authors contributed equally: Jason A. Trubiano, Claire L. Gordon, Thi H. O. Nguyen, Katherine Kedzierska. ✉email: trubianoj@unimelb.edu.au; claire.gordon@unimelb.edu.au; tho.nguyen@unimelb.edu.au; kkedz@unimelb.edu.au

