## [Peer Review File · Nature Communications]

SARS-CoV-2 infection results in immune responses in the respiratory tract and peripheral blood that suggest mechanisms of disease severityREVIEWER COMMENTS

Reviewer #1 (Remarks to the Author):

In this manuscript, Wuji Zhang and colleagues analyzed cellular, humoral, and cytokine responses in paired blood and respiratory tract samples from COVID-19 patients and non-COVID-19 controls. They found discordant cytokine level, higher IgM and IgG seroconversion, and higher frequencies of neutrophils and effector memory (EM)-like CD4+ and CD8+ T-cells in respiratory samples compared to paired blood. They also found that dexamethasone and/or remdesivir therapy did not reduce humoral responses in COVID-19 patients. They revealed the differences in innate and adaptive immune responses between respiratory samples and blood. Their data are descriptive and providing limited insights into mechanistically understanding of disease pathogenesis.

Specific points:

1. It is confusing that cytokine profiles varied across COVID-19 respiratory samples, but no significant differences were found in their cellular immunity (line 328-329). Any correlation of the cellular, humoral, and cytokine responses between respiratory and blood samples were observed? In addition, did any of these factors correlate with disease severity.

2. The time point of samples accessed in the Fig. 2 to 6 should be clarified in detail, as cellular, humoral, and cytokine responses varied over time of disease process.

3. In Fig.2, IL-6, IL-8, and MCP-1 were most prevalent in respiratory samples in both COVID-19 patients and non-COVID-19 patients. Were these non-COVID-19 patient infected with other respiratory pathogens?

4. The authors did not indicate sample types accessed in Fig.7 and Fig.8, please clarify. In addition, are there any differences between respiratory samples and plasma from patients with/without dexamethasone treatment?

5. Most of the published work are based on blood samples since blood samples are easier to acquire and process . Is it possible to correlate or predict inflammatory and immune responses in respiratory system from the blood samples?

6. The data and manuscript are not well organized and hard to follow.

Reviewer #2 (Remarks to the Author):

The manuscript is very interesting as it explores respiratory immune response which is an understudied topic. The methods for the work are well detailed.

However, as mentioned by authors, the study has a lot of limitations and personally, the work need major revision

In the introduction section I suggest to remove the paragraph from line 124 to line 132 which is not an introduction but a summary discussion of the data.

The study population is not clearly explained across the manuscript. For example line 145: 12 ward patients and six ICU patients were on dexamethasone treatment, while 8 ward patients and 10 ICU patients were on dexamethasone (with/without remdesivir) treatment.

Moreover, the sample collection is not uniform and this dramatically affects the strength of the study. For example : line 150: 97% of blood samples were collected at hospital admission (Visit 1/V1) or discharge (V7; n=62 and n=35 out of 99 blood samples, respectively), with others collected during hospitalization (V2:ICU admission, V3:acute respiratory distress syndrome/cytokine release syndrome diagnosis, V5:24-48 hours post drug therapy, V6:7-14 days post drug therapy). I think this is an important point to consider especially when debating the role of corticosteroid therapy

Reviewer #3 (Remarks to the Author):

The main hypothesis behind this study is interesting: does measuring immune mediators, humoral and cellular responses from respiratory samples add value compared with blood, the area which most studies have focused on.

The immunological investigations presented in the manuscript are extensive and reported methodologies are sound. However, unfortunately the wealth of information provided lacks focus, many of the findings reported are well understood and do not add significantly to the field, while the actual utility of measuring immune responses in respiratory samples remains unclear.

A fundamental issue is the small number of respiratory samples available for analysis (only 10 individuals with COVID-19) and their heterogeneous nature - which includes a mixture of ETT aspirates and sputum (plus one pleural fluid!).

- Clearly the pleural fluid is not relevant to the study hypothesis: it is not from the airways
- No data is presented to determine the quality of the sputum/ETTA samples - both have a high propensity to be contaminated e.g. from the oropharynx, or reflecting sampling of the large and not lower airways
- This heterogeneity in the samples and sampling is a very important confounder and is likely to partially explain the heterogeneity in observed results between patients

Secondly, the authors provide a descriptive overview of the differences in inflammatory response between blood and respiratory samples, but provide little analysis of what value this adds

- For example the statement in lines 221-222 is a good description of what the study authors hope to achieve, however no data is presented to indicate that measuring inflammatory mediators in sputum is predictive of outcomes, treatment effects etc.
- Where analysis is presented looking at outcomes it of limited utility for example in paragraph lines 173-184 the findings are entirely as expected, while in paragraph lines 357-371 the conclusions are flawed as they rely on analysing samples at discharge and not during the illness

Thirdly it is unclear how the 6 individuals without COVID were chosen and what value they add to the study

- inclusion/exclusion criteria for study enrolment are not presented
- diagnosis of the 6 individuals admitted to ICU without COVID is not presented
- it is not really clear how these 6 individuals improve our understanding of the inflammatory response in COVID, and comparing SARS-CoV-2 titres in this group with those who do have COVID is facile (e.g. lines 228-229, 261-262)

RESPONSES TO REVIEWERS' COMMENTS

We immensely thank the Reviewers for their comments and insightful suggestions, which allowed us to greatly improve the manuscript.

Following the comments from the Reviewers, we have substantially revised our manuscript, extended our respiratory cohort and performed analyses on the additional matched respiratory and plasma samples.

In short, we have increased the number of respiratory samples from 14 in the original manuscript to 41 in the revised version. These 41 COVID-19 samples include 15 endotracheal aspirates (ETA; from 11 patients), 20 sputum (from 18 patients) and 6 bronchoalveolar lavage (BAL; from 6 patients). The respiratory samples were obtained from 33 COVID-19 patients, with 34 paired blood samples.

The additional samples include all possible respiratory samples available to us via our collaborative links with 5 major hospitals in Melbourne and Sydney. We have collected a further 28 respiratory samples (5 ETA, 17 sputum, 6 BAL samples) and 22 paired blood samples from 24 COVID-19 patients. Overall, this is a substantial increase from 14 respiratory samples (10 ETA samples, 3 sputum samples and 1 pleural sample) in the original manuscript.

We have performed additional analyses to assess antibody levels, their neutralising activity as well as cytokine/chemokine levels, and provided insights into how these data correlate with disease severity. Thus, in our revised manuscript, the substantial numbers of respiratory samples now allow sufficient statistical analyses to be conducted across different types of respiratory specimens to account for the heterogenous nature of the specimens.

Reviewer #1 (Remarks to the Author):

Specific points:

1. It is confusing that cytokine profiles varied across COVID-19 respiratory samples, but no significant differences were found in their cellular immunity (line 328-329). Any correlation of the cellular, humoral, and cytokine responses between respiratory and blood samples were observed? In addition, did any of these factors correlate with disease severity.

We thank the Reviewer for this comment and the opportunity to clarify this point. In the revised version of the manuscript, we have extended our cohort to 41 respiratory samples and performed analyses on the additional 28 respiratory samples. Our analyses across 41 respiratory samples show more uniform cytokine/chemokine production across respiratory samples. This is exemplified by markedly increased levels of IL-6, MCP-1 and IL-8 in respiratory samples compared to matched plasma in all but 5 COVID-19 patients.

We have modified Results and Figure 2 to incorporate the new data (page 7):

“Amongst the COVID-19 patients, greatly elevated levels of inflammatory cytokines/chemokines were detected in respiratory samples across ETA, sputum and BAL specimens, with concentrations being 160x (MCP-1), 90x (IL-6) and 110x (IL-8) higher than in plasma (Fig. 2a). While IL-18 dominated in plasma, IL-6, IL-8 and MCP-1 were

most prevalent in respiratory samples in patients with high cytokines/chemokines (Fig. 2a, Supplementary Fig. 1a). MCP-1, IL-6 and IL-8 were significant higher in sputum and BAL than in plasma ($p < 0.0001$ and $p = 0.0476$; higher median also observed in ETA although not significant), IFN- γ , IL-12p70, IL-17A, IL-23 and IL-33 were significantly higher in plasma than in respiratory samples ($p < 0.0001$ - $p = 0.0484$; Fig. 2b). IL-1 β and IL-18 were also higher in sputum but not in ETA or BAL than in plasma ($p < 0.0001$ and $p = 0.0023$ respectively; Fig. 2b). In contrast, concentrations of IFN- $\alpha 2$, IL-10 and TNF were comparable across respiratory and plasma specimens, while sIL-6R α was lower in respiratory specimens than in plasma (Fig. 2b, Supplementary Fig. 1b). Furthermore, we found significant correlations between respiratory samples and plasma with respect to IL-1 β , MCP-1, IFN $\alpha 2$, IFN γ , IL-8 and IL-10 (Fig. 2c). It is important to note that the 13 cytokines/chemokines measured were not detected in ETA or sputum samples of five COVID-19 ICU patients (#002, #003, #004, #011, #013), while high IL-18 levels were detected in plasma of patients #002, #003, #004 and #013 (Supplementary Fig. 1a), demonstrating that cytokine levels can vary across both COVID-19 patients in respiratory samples and between paired respiratory and plasma samples.

High cytokine levels were detected in ETA samples from only 2 out of 11 COVID-19 patients (#26 and #49) (Fig. 2d). In sputum samples, however, this was seen in 10 out of 18 patients (#71, #72, #73, #74, #74, #76, #80, #81, #82, #84). Our data therefore suggest that sputum potentially represents the most desirable specimen that reflects high inflammatory milieu at the primary site of SARS-CoV-2 infection. Conversely, the majority of patients displayed elevated cytokine/chemokine levels in plasma.”

Following the Reviewer’s comment, we have also added Fig.2c to show how cytokine levels correlate between respiratory and paired plasma samples:

We have also performed in-depth correlation analyses between immune parameters and disease outcome, including correlations between cytokine/chemokine production,

antibodies, disease severity and age. We found that IFN- α 2 and IL-12p70 levels in ETA negatively correlated with days of hospital stay (Fig. 6ab). In sputum, sVNT inhibition activity negatively correlated with days of hospital stay, while it positively correlated with levels of MCP-1, IL-6, IL-8 and IL-10 (Fig. 6cd).

We have described the above data in the Result section (page 11):

“IFN- α 2 and IL-12p70 levels in ETA and RBD neutralizing activity in sputum negatively correlate with days of hospital stay

To understand associations between clinical features and serological responses in the respiratory specimens, correlations between clinical data (age, weight, height, BMI, days post disease onset, days of hospital stay) and serological features (cytokines and chemokines, sIL-6R α , ADAMTS4, anti-RBD IgM, IgG, IgA, sVNT inhibition) were performed for ETA and sputum samples separately (Fig. 6a-d). IFN- α 2 and IL-12p70 levels in ETA negatively correlated with days of hospital stay, albeit low levels of IL-12p70 in respiratory samples (Fig. 6a, b). In sputum, sVNT inhibition activity negatively correlated with days of hospital stay, while it positively correlated with levels of MCP-1, IL-6, IL-8 and IL-10 (Fig. 6cd).”

In addition, we have included Fig 6a-d which shows an overall correlation of respiratory cytokine levels, antibody levels and clinical features:

Moreover, we found overall IgM, IgG and sVNT levels correlated between respiratory samples and plasma (Fig. 3f; page 9).

We were unable to analyse correlations with cellular events as the majority of the additional samples comprised only of supernatant.

2. The time point of samples accessed in the Fig. 2 to 6 should be clarified in detail, as cellular, humoral, and cytokine responses varied over time of disease process.

We have ensured that we clarified the time of samples assessed in Figures 2 to 6. In the revised version of the manuscript, we have additionally provided information on ‘days post disease onset’ throughout.

3. In Fig.2, IL-6, IL-8, and MCP-1 were most prevalent in respiratory samples in both COVID-19 patients and non-COVID-19 patients. Were these non-COVID-19 patient infected with other respiratory pathogens?

Yes, these were non-COVID-19 patients with other diseases. Following queries from Reviewer 1 and Reviewer 3, we have removed these non-COVID-19 patients from cytokine analyses so as to not distract from the main focus on immune responses in COVID-19 respiratory samples. They were however retained as negative controls for antibody responses, since they were PCR-negative and had low respiratory and plasma SARS-CoV-2-specific antibody levels.

4. The authors did not indicate sample types accessed in Fig.7 and Fig.8, please clarify. In addition, are there any differences between respiratory samples and plasma from patients with/without dexamethasone treatment?

In the revised version of the manuscript and figures, we have indicated the sample type assessed in Fig 7 and 8, which is peripheral blood.

We have added a label for the sample type in Fig 7 and 8:

Unfortunately, we could not perform analyses to measure differences between respiratory samples and plasma from patients with/without dexamethasone as 28 out of 33 patients with respiratory samples collected were given dexamethasone, while the drug treatment for 2 patients was not known.

5. *Most of the published work are based on blood samples since blood samples are easier to acquire and process. Is it possible to correlate or predict inflammatory and immune responses in respiratory system from the blood samples?*

As we described above, with the extended cohort of respiratory samples (n=41), we found that amongst the COVID-19 patients, greatly elevated levels of inflammatory cytokines/chemokines were detected in respiratory samples across ETA, sputum and BAL specimens, with concentrations being 160x (MCP-1), 90x (IL-6) and 110x (IL-8) higher than in plasma. While IL-18 dominated in plasma, IL-6, IL-8 and MCP-1 were most prevalent in respiratory samples in patients with high cytokines/chemokines (Fig. 2a, Supplementary Fig. 1a). We found significant correlations between respiratory samples and plasma with respect to IL-1 β , MCP-1, IFN- α 2, IFN- γ , IL-8 and IL-10 (Fig. 2c). Especially, concentrations of IFN- α 2, IL-10 and TNF were comparable across respiratory and plasma specimens (Fig. 2b, Supplementary Fig. 1b).

6. *The data and manuscript are not well organized and hard to follow.*

We have re-organised the data in the manuscript, so our findings are now easier to follow. The major modifications are:

Fig 1 (cohort introduction figure)

1a: We have included a new cohort flowchart to clarify our cohort and split respiratory and blood analyses.

1b, c, e: We have modified the figures to include additional data from extended COVID-19 patient cohort.

Fig 2 (cytokine figure)

Non-COVID-19 samples and 1 pleural fluid were removed from cytokine analyses

2a: We have re-plotted the cytokine data for pooled respiratory and paired plasma samples, as well as separated the data according to respiratory specimen types.

2b: We have analysed individual cytokine levels by separating respiratory specimens into ETAs, sputum and BAL. New statistical analyses between paired plasma and respiratory samples are shown.

2c: New correlation analyses between respiratory and paired plasma collected at similar times are shown and separated by respiratory specimen types.

2d: Additional respiratory and plasma samples were included in our heatmap analyses.

Fig 3 (antibody figure)

3b(i-ii): Data from additional respiratory and plasma samples were added.

3b(iii): New comparisons between plasma and respiratory samples separated according to their specimen type are shown.

3c, d, e: Data from additional respiratory and plasma samples were added.

3f: New correlation analyses between respiratory and paired plasma collected at similar times and separated by respiratory specimen types, were performed.

Fig 6 (correlation of respiratory features figure)

6a-d. New correlation analyses between age, days post disease onset, days of hospital stay, respiratory cytokine and antibody levels were performed, separated by respiratory specimen types.

Reviewer #2 (Remarks to the Author):

1. The manuscript is very interesting as it explores respiratory immune response which is an understudied topic. The methods for the work are well detailed. However, as mentioned by authors, the study has a lot of limitations and personally, the work need major revision. In the introduction section I suggest to remove the paragraph from line 124 to line 132 which is not an introduction but a summary discussion of the data.

Following the Reviewer's comment, we have removed the paragraph from line 124 to line 132.

2. The study population is not clearly explained across the manuscript. For example line 145: 12 ward patients and six ICU patients were on dexamethasone treatment, while 8 ward patients and 10 ICU patients were on dexamethasone (with/without remdesivir) treatment.

In the revised version of the manuscript, we have re-written the cohort section separated for analyses of respiratory and blood samples to ensure its clarity (page 6):

“To define immune responses to SARS-CoV-2 in the respiratory tract, we obtained 41 respiratory samples (15 endotracheal aspirates (ETA; from 11 patients), 20 sputum (from 18 patients), 6 bronchoalveolar lavage (BAL; from 6 patients). Respiratory samples were collected from 33 PCR-positive COVID-19 patients from whom we also collected 34 paired blood samples (Fig. 1a, Supplementary Table 1 and Supplementary Table 2). Three COVID-19 patients were admitted to the ward while 30 patients were in the ICU (Fig. 1c; Supplementary Table 1). The median age of COVID-19 patients from whom we obtained respiratory samples was 55 years (range 25-76) and 33.3% were females (Supplementary Table 1). When feasible, blood was collected on hospital admission, during hospital stay and on hospital discharge. No significant differences were found between time of respiratory specimen and matched blood samples collected at the closest time-point ($p=0.89$; Fig. 1b).

To determine the effects of dexamethasone, an anti-inflammatory corticosteroid, taken alone or in combination with the anti-viral drug remdesivir on immune responses in blood, we recruited 57 COVID-19 patients (42 ward patients and 15 ICU patients) from whom we obtained 86 blood samples, with a median age of 58 (range 22-90) and 49.1% of females (Fig. 1a, Supplementary Table 1, Supplementary Table 2).”

We have also included a schematic depicting for our cohort and sample distribution in Figure 1a.

3. Moreover, the sample collection is not uniform and this dramatically affects the strength of the study. For example: line 150: 97% of blood samples were collected at hospital admission (Visit 1/V1) or discharge (V7; $n=62$ and $n=35$ out of 99 blood samples, respectively), with others collected during hospitalization (V2:ICU admission, V3:acute respiratory distress syndrome/cytokine release syndrome diagnosis, V5:24-48 hours post drug therapy, V6:7-14 days post drug therapy). I think this is an important point to consider especially when debating the role of corticosteroid therapy.

We thank the Reviewer for pointing this out. As mentioned above, we have modified the information on our cohort to clarify the sample collection. We hope the Reviewer can appreciate that due to the nature of clinical sampling in severely-ill COVID-19 patients, these individuals could not be bled on exactly the same days after disease onset. In the majority of patient cases, whenever it was feasible, we have collected blood samples on hospital admission and hospital discharge, with additional samples collected during the hospital stay and closely matched to the respiratory sample collection times. Furthermore, for clarity, we have made changes throughout the manuscript and refer time points to days post disease onset. Analyses of immune responses in the blood were analysed by hospital admission (V1) and hospital discharge (V7).

Reviewer #3 (Remarks to the Author):

The main hypothesis behind this study is interesting: does measuring immune mediators, humoral and cellular responses from respiratory samples add value compared with blood, the area which most studies have focused on.

1. The immunological investigations presented in the manuscript are extensive and reported methodologies are sound. However, unfortunately the wealth of information provided lacks focus, many of the findings reported are well understood and do not add significantly to the field, while the actually utility of measuring immune responses in respiratory samples remains unclear.

Following the Reviewer's comment, we have modified the manuscript to focus on our findings and clarify their novelty. The novel aspects of our study is related to the breadth of immune features (a total of 382 immune features, including 315 multiplex antibody features) analysed in matched respiratory and plasma samples. We also provide data on

correlations between immune parameters in respiratory samples with disease severity (i.e. duration of hospital stay) and other physiological factors, including age, weight, height, BMI and days post disease onset. Our findings revealed that IFN- α 2 and IL-12p70 levels in ETA negatively correlated with days of hospital stay (Fig. 6a, b). In sputum, sVNT inhibition activity negatively correlated with days of hospital stay, while it positively correlated with levels of MCP-1, IL-6, IL-8, and IL-10 (Fig. 6cd).

We are also the first to demonstrate that cytokine levels can vary across both COVID-19 patients in respiratory samples and between paired respiratory and plasma samples. In general, respiratory samples from COVID-19 patients showed significantly higher levels of several cytokines in comparison to plasma, including MCP-1, IL-6 and IL-8 (Fig. 2b). While MCP-1, IL-6, and IL-8 were significantly higher in sputum and BAL than in plasma ($p < 0.0001$ - $p = 0.0476$; higher median also observed in ETA although not significant), IFN- γ , IL-12p70, IL-17A, IL-23, and IL-33 were significantly higher in plasma than in respiratory samples ($p < 0.0001$ - $p = 0.0484$; Fig. 2b). IL-1 β and IL-18 were also higher in sputum but not in ETA or BAL than in plasma ($p < 0.0001$ and $p = 0.0023$ respectively; Fig. 2b). In contrast, concentrations of IFN- α 2, IL-10 and TNF were comparable across respiratory and plasma samples, while sIL-6R α was lower in respiratory specimens than in plasma (Fig. 2b, Supplementary Fig. 1b). We found significant correlations between respiratory samples and plasma with respect to IL-1 β , MCP-1, IFN α 2, IFN γ , IL-8 and IL-10 correlated (Fig. 2c).

Following the Reviewer's comment, we have also re-written the Abstract to emphasize the novel aspects of our study.

2. A fundamental issue is the small number of respiratory samples available for analysis (only 10 individuals with COVID-19) and their heterogenous nature - which includes a mixture of ETT aspirates and sputum (plus one pleural fluid!).

We agree with the Reviewer that the sample size was small in our original submission, but these were the only samples we could access in 2020 from the second wave of SARS-CoV-2 in Melbourne. However, as mentioned above, we have now substantially increased our cohort to 41 respiratory samples, which include 15 endotracheal aspirates (ETA; from 11 patients), 20 sputum (from 18 patients) and 6 bronchoalveolar lavage (BAL; from 6 patients). The respiratory samples were obtained from 33 COVID-19 patients, with 34 paired blood samples. The additional samples include all the respiratory samples we could obtain via our collaborative links with 5 major hospitals in Melbourne and Sydney.

As we described above, our new analyses showed correlations between levels of cytokines/chemokines in respiratory samples, antibodies and disease outcome.

3. Clearly the pleural fluid is not relevant to the study hypothesis: it is not from the airways

We agree with the Reviewer's point and have removed the pleural fluid from our analyses, with the exception of the multidimensional system serology assay and Flow Self-Organizing Map, as the latter analyses would need to be re-run totally, thus causing substantial delays.

4. No data is presented to determine the quality of the sputum/ETTA samples - both have a

high propensity to be contaminated e.g. from the oropharynx, or reflecting sampling of the large and not lower airways

ETA and sputum samples were collected by highly trained clinicians and research nurses in ICU settings. Drs Claire Gordon and Olivia Smibert met with the head of ICU to develop a specific protocol for the collection of these samples that was standardized, consistent and reproducible. Dr Smibert supervised the collection of every samples and can confirm that the protocol was strictly and consistently followed. Dr Smibert also took responsibility for transportation of samples from ICU to the ID laboratory to ensure a consistent transport chain from the point of aspiration, processing and transfer to the Doherty Institute on ice.

For clarity, the protocol used in the Austin hospital was:

1. Prior to sample collection, the suction catheter from the endotracheal tube (ETT) from each patient was replaced with a new sterile suction catheter.
2. 10ml of sterile saline was administered down the ETT via the suction catheter and allowed to sit for up to 5 seconds (depending on whether a cough was elicited and the stillness of the patient etc.)
3. A sterile respiratory sample trap was attached to the suction catheter of the ETT and sample aspirated directly into the respiratory trap and placed on ice
4. The sample was transported to the Austin ID laboratory where 500uL was split for storage in DNA/RNA Shield (and the remainder transported immediately to the Doherty Institute).

Furthermore, to verify the quality, we have tabulated the immunological readouts of the 6 respiratory samples with undetectable cytokine/chemokine levels. As shown in Supplementary Table 6, we found detectable levels of soluble IL-6 receptor, ADAMTS4 and antibodies/neutralizing activities in these respiratory samples, reflecting their high quality and integrity.

Sample	sIL6Ra	ADAMTS4	IgM	IgG	IgA	% inhibition sVNT
002 Sputum	2.381	15.625	2.645	1.239	4.101	3.448
003 ETA	2.970	1264.532	3.445	3.686	3.618	28.966
004 ETA d7	2.360	15.625	0.000	1.646	1.772	3.448
004 ETA d14	33.553	15.625	3.796	4.321	4.081	48.828
011 Sputum	4.404	15.625	2.456	2.818	2.366	0.690
013 ETA	6.613	15.625	1.296	2.209	2.622	0.000

5. This heterogeneity in the samples and sampling is a very important confounder and is likely to partially explain the heterogeneity in observed results between patients

As specified above, we have now substantially increased the numbers of respiratory samples to 41 samples (15 ETAs, 20 sputum and 6 BAL samples). We have performed additional analyses to assess antibody levels, their neutralising activity as well as cytokine/chemokine levels, and performed correlation analyses with disease severity. Thus, in our revised manuscript, we now have substantial numbers of respiratory samples to ensure statistical power to split the analyses across different types of respiratory samples to reduce the heterogeneity in our data.

6. Secondly, the authors provide a descriptive overview of the differences in inflammatory response between blood and respiratory samples, but provide little analysis of what value this adds

As we have extended our respiratory cohort, we performed additional analyses with statistics for the data in our manuscript. These are indicated in Fig 1, 2, 3, 4, 5 and 6.

7. For example the statement in lines 221-222 is a good description of what the study authors hope to achieve, however no data is presented to indicate that measuring inflammatory mediators in sputum is predictive of outcomes, treatment effects etc.

Having analysed a larger respiratory cohort, our data on inflammatory mediators in respiratory samples show that IFN- α 2 and IL-12p70 levels in ETA negatively correlated with days of hospital stay, albeit there were low levels of IL-12p70 in respiratory samples (Fig. 6ab). In sputum, sVNT inhibition activity negatively correlated with days of hospital stay, but positively correlated with levels of MCP-1, IL-6, IL-8 and IL-10 (Fig. 6cd).

8. Where analysis is presented looking at outcomes it of limited utility for example in paragraph lines 173-184 the findings are entirely as expected, while in paragraph lines 357-371 the conclusions are flawed as they rely on analysing samples at discharge and not during the illness

We have now revised this section. Lines 357-361 revealed different antibody responses in plasma of Mild/Moderate and Severe/Critical patients grouped by NIH scores. Regarding disease outcomes, PLSDA revealed that patients prior to dexamethasone therapy had higher antibodies in plasma against the NP of human coronavirus OC43 rather than SARS-CoV-2 at hospital admission, providing insights into potential needs of drug treatment based on patient's antibody responses at hospital admission (Fig. 8c).

9. Thirdly it is unclear how the 6 individuals without COVID were chosen and what value they add to the study

- inclusion/exclusion criteria for study enrolment are not presented
- diagnosis of the 6 individuals admitted to ICU without COVID is not presented
- it is not really clear how these 6 individuals improve our understanding of the inflammatory response in COVID, and comparing SARS-CoV-2 titres in this group with those who do have COVID is facile (e.g. lines 228-229, 261-262)

Following comments from both Reviewer 1 and Reviewer 3, we have removed the non-COVID-19 respiratory samples from cytokine analyses so as not to distract the reader from the main message of the manuscript. They were kept as negative controls for antibody responses since they were PCR negative and had low respiratory and plasma SARS-CoV-2-specific antibody levels.

REVIEWERS' COMMENTS

Reviewer #2 (Remarks to the Author):

I am grateful to the authors for their revisions. The authors have adequately addressed all my queries. In general, the paper has evolved to a reasonable contribution to the scientific community.

Reviewer #3 (Remarks to the Author):

The authors have provided an intriguing manuscript which has been refocused compared to the original submission, and is much the better for it.

The inclusion of additional respiratory and blood samples clearly strengthens the findings - and the intention of comparing cellular, humoral and inflammatory responses between blood and lower respiratory samples is sound.

I believe there is a good manuscript here, but to get to it will require extensive further editing to remove extraneous details:

The first subsection of the results is titled 'ICU admission associated with higher NIH severity score, oxygen therapy, drug treatment and weight'. This whole section is true, but obvious and unnecessary (much of the associated figures can also be removed)

The subsequent section exploring cytokine levels is OK, but the distinction between BAL/ETA/sputum shows a lack of clinical understanding. Fundamentally these samples are either representative of the lower airways, or they are not (and BAL would be expected to be more representative than ETA, and that more so than sputum). Presenting the findings of each separately is unnecessary - this could be moved to supplementary. I suspect the finding that sputum had higher inflammatory levels reflects dilution that occurs during lavaging for BAL collection (and sometimes endotracheal tube aspiration). So the conclusion to this paragraph may be right, but the clinical reasoning is lacking. Further the authors conclude 'hospitalized/ICU COVID-19 patients should be monitored for inflammation in airways... to understand disease severity and potential benefits of immunomodulatory treatments' - this is their hypothesis, but they present no data to back this up, beyond description of the differences. Indeed this study is titled 'Immune responses in COVID-19 respiratory tract and blood reveal mechanisms of disease severity' - this study is primarily descriptive, and reveals very little about mechanisms.

In the next 2 sections the authors have persisted with comparing SARS-CoV-2 antibody levels between patients with and without COVID, which is irrelevant. As is the PLSDA and as are the antibody levels against seasonal coronaviruses. And pleural fluid should not be retained in Figure 4 for convenience. Removing this and the associated figures will further help to focus the paper.

The next section 'Increasing cellular infiltrates in respiratory specimens during disease progression' is fine except for again including samples from non-COVID patients.

The section 'COVID-19 patients with higher NIH scores had more robust humoral immune responses in blood' is, as the authors note 'unsurprising' and this whole section can be removed without detracting from the point of the manuscript.

The final section looking at the effect of dexamethasone and remdesivir on immune responses is a worthwhile endeavor, but needs careful study to control for confounding - the timing of administration in the disease course, the timing of sample collection, other medications which may be administered (e.g. tocilizumab, baricitinib), complications such as secondary bacterial infections. This section exists as an addendum which can also be removed.

Finally there are numerous instances throughout the manuscript (from the very first sentence in the abstract) where unfortunately the language needs considerable polishing.

RESPONSES TO REVIEWERS' COMMENTS

Reviewer #2 (Remarks to the Author):

I am grateful to the authors for their revisions. The authors have adequately addressed all my queries.

In general, the paper has evolved to a reasonable contribution to the scientific community.

We immensely thank the Reviewers for their comments.

Reviewer #3 (Remarks to the Author):

The authors have provided an intriguing manuscript which has been refocused compared to the original submission, and is much the better for it. The inclusion of additional respiratory and blood samples clearly strengthens the findings - and the intention of comparing cellular, humoral and inflammatory responses between blood and lower respiratory samples is sound.

We thank the Reviewer for the comments.

I believe there is a good manuscript here, but to get to it will require extensive further editing to remove extraneous details:

1. The first subsection of the results is titled 'ICU admission associated with higher NIH severity score, oxygen therapy, drug treatment and weight'. This whole section is true, but obvious and unnecessary (much of the associated figures can also be removed)

We believe the figures provide necessary information about the cohort, and so we would like to keep them if that is okay with the Editor.

2. The subsequent section exploring cytokine levels is OK, but the distinction between BAL/ETA/sputum shows a lack of clinical understanding. Fundamentally these samples are either representative of the lower airways, or they are not (and BAL would be expected to be more representative than ETA, and that more so than sputum). Presenting the findings of each separately is unnecessary - this could be moved to supplementary. I suspect the finding that sputum had higher inflammatory levels reflects dilution that occurs during lavaging for BAL collection (and sometimes endotracheal tube aspiration). So the conclusion to this paragraph may be right, but the clinical reasoning is lacking.

Given the heterogeneity of the samples, as mentioned by Reviewer #3's previous comments, we believe it was necessary to present the findings separately for each sample type and thereby reducing the heterogeneity in our data, especially after having substantially increased the numbers of respiratory samples to 41 samples (15 ETAs, 20 sputum and 6 BAL samples).

3. Further the authors conclude 'hospitalized/ICU COVID-19 patients should be monitored for inflammation in airways... to understand disease severity and potential benefits of immunomodulatory treatments' - this is their hypothesis, but they present no data to back this up, beyond description of the differences. Indeed this study is titled 'Immune responses in

COVID-19 respiratory tract and blood reveal mechanisms of disease severity' - this study is primarily descriptive, and reveals very little about mechanisms.

Following the Reviewer's comment, we have changed the title to the one suggested by the Editor, which now reads: 'SARS-CoV-2 infection results in immune responses in the respiratory tract and peripheral blood that suggest mechanisms of disease severity'.

4. In the next 2 sections the authors have persisted with comparing SARS-CoV-2 antibody levels between patients with and without COVID, which is irrelevant. As is the PLSDA and as are the antibody levels against seasonal coronaviruses. And pleural fluid should not be retained in Figure 4 for convenience. Removing this and the associated figures will further help to focus the paper.

We believe the analyses of respiratory antibody levels reveal insights into humoral immune responses in COVID-19 patients in comparison to non-COVID patients, hence we would like to keep the data if that is okay with the Editor.

5. The next section 'Increasing cellular infiltrates in respiratory specimens during disease progression' is fine except for again including samples from non-COVID patients.

Following the Reviewers' comments from the first round, we have moved the non-COVID-19 cellular data into Supplementary Figure 4. We would like to keep them in the Supplementary figure if that is okay with the Editor.

6. The section 'COVID-19 patients with higher NIH scores had more robust humoral immune responses in blood' is, as the authors note 'unsurprising' and this whole section can be removed without detracting from the point of the manuscript.

The final section looking at the effect of dexamethasone and remdesivir on immune responses is a worthwhile endeavor, but needs careful study to control for confounding - the timing of administration in the disease course, the timing of sample collection, other medications which may be administered (e.g. tocilizumab, baricitnib), complications such as secondary bacterial infections. This section exists as an addendum which can also be removed.

We believe the data in Figures 7 and 8 provide new findings about the impact of drug treatment (dexamethasone and remdesivir) on immune responses within the blood, we would like to keep them if that is okay with the Editor.

7. Finally there are numerous instances throughout the manuscript (from the very first sentence in the abstract) where unfortunately the language needs considerable polishing.

Following the Reviewer's comment, our manuscript was thoroughly reviewed by Dr. Bridie Clemens, who is highly proficient in English grammar.